



# Assessing the value of BGC Argo profiles versus ocean colour observations for biogeochemical model optimization in the Gulf of Mexico

Bin Wang[1], Katja Fennel[1], Liuqian Yu[1,2], Christopher Gordon[1]

[1]Department of Oceanography, Dalhousie University, Halifax, Nova Scotia, Canada

[2]Department of Mathematics, The Hong Kong University of Science and Technology, Kowloon, Hong Kong

*Correspondence to*: Bin Wang (Bin.Wang@dal.ca)

**Abstract**. Biogeochemical ocean models are useful tools subject to uncertainties arising from simplifications, inaccurate parameterization of processes, and poorly known model parameters. Parameter optimization is a standard method for addressing the latter but typically cannot constrain all biogeochemical parameters because of insufficient observations. Here we assess the trade-offs between satellite observations of ocean colour and biogeochemical (BGC) Argo profiles, and the benefits of combining both observation types, for optimizing biogeochemical parameters in a model of the Gulf of Mexico. A suite of optimization experiments is carried out using different combinations of satellite chlorophyll and profile measurements of chlorophyll, phytoplankton biomass, and particulate organic carbon (POC) from autonomous floats. As parameter optimization in 3D models is computationally expensive, we optimize the parameters in a 1D model version, and then perform 3D simulations using these parameters. We show first that the use of parameters optimized in 1D improves the skill of the 3D model. Parameters that are only optimized with respect to surface chlorophyll cannot reproduce subsurface distributions of biological fields. Adding profiles of chlorophyll in the parameter optimization yields significant improvements for surface and subsurface chlorophyll but does not accurately capture subsurface phytoplankton and POC distributions because the parameter for the maximum ratio of chlorophyll to phytoplankton carbon is not well constrained in that case. Using all available observations leads to significant improvements of both observed (chlorophyll, phytoplankton, and POC) and unobserved variables (e.g. primary production). Our results highlight the significant benefits of BGC Argo measurements for biogeochemical parameter optimization and model calibration.



## 1. Introduction

Oceanic primary production forms the basis of the marine food web and fuels the biological pump, which contributes to the sequestration of atmospheric $CO_2$ in the ocean's interior thus mitigating global warming. An accurate quantification of primary production and biological carbon export is therefore important for our understanding of the marine carbon cycle and for predicting how carbon cycling and marine ecosystems will interact with climate change.

Direct observations of primary production and export flux are relatively sparse because of the cost and effort involved in measuring these fluxes. Numerical models can complement sparse observations. Well validated and calibrated models are useful tools for hindcasting and nowcasting past and present biogeochemical fluxes and are the most common tool for projecting future changes.

In recent years, many biogeochemical models with different complexities, ranging from three to more than thirty biological state variables, have been developed to study ocean biogeochemical processes. Regardless of their complexities, the performance of these models is highly dependent on the appropriate choice of model parameter values (e.g. maximum growth, grazing and mortality rates), most of which are poorly known. Some parameter choices are informed by laboratory experiments (e.g. light and nutrient dependence of phytoplankton growth), although isolated cultures in the lab may not be representative of the behavior of natural communities. Other parameters cannot be directly determined (e.g. mortality rates). Choosing appropriate parameter values becomes more challenging as model complexity grows because the number of parameters increases exponentially with the number of state variables (Denman, 2003).

A standard method for addressing the problem of poorly known model parameters is parameter optimization, a process by which the misfit between model results and available observations is minimized by iteratively varying parameters (Matear, 1995; Prunet et al., 1996a, 1996b; Fennel et al., 2001; Friedrichs et al., 2007; Kuhn et al., 2015, 2018). However, even formal optimization typically cannot constrain all biogeochemical parameters (i.e. provide optimal parameter estimates with relatively small uncertainties) because of insufficient information in the available observations (Matear, 1995; Fennel et al., 2001; Ward et al., 2010; Bagniewski et al., 2011). For example, Matear (1995) used a so-





called simulated annealing algorithm to optimize three different ecosystem models and found that even
for the simplest nutrient-phytoplankton-zooplankton model, not all independent parameters could be
constrained well, leaving the others with large uncertainty ranges. A more recent study reported that the
lack of zooplankton observations led to poor accuracy of the optimized zooplankton-related parameters
when using a suite of Lagrangian-based observations during the North Atlantic spring bloom (Bagniewski
et al., 2011). A broader suite of observation types should be favourable to parameter optimization although
complications can arise. For example, when optimizing a suite of 1D models for the Mid-Atlantic Bight,
the use of satellite POC observations in addition to satellite chlorophyll did not yield further
improvements in model-data fit but degraded the representation of chlorophyll (Xiao and Friedrichs,
2014a).

64         Typically surface ocean chlorophyll from satellite is the main source of observations for model

validation (e.g. Doney et al., 2009; Gomez et al., 2018; Lehmann et al., 2009) and parameter optimization
(Prunet et al., 1996a; Xiao and Friedrichs, 2014a, 2014b), supplemented by other observation types as
available. However, satellites only see the ocean surface and do not resolve the vertical distribution of
chlorophyll. This is especially problematic in oligotrophic regions where the maximum chlorophyll
concentration (referred as the deep chlorophyll maximum, DCM) is pronounced near the base of the
euphotic zone because of photoacclimation (Cullen, 2015; Fennel and Boss, 2003). In addition, although
chlorophyll has long been used as a proxy of phytoplankton biomass and to estimate primary production
based on some assumptions (Behrenfeld and Falkowski, 1997), it is not a direct measure of carbon-based
phytoplankton biomass. The ratio of chlorophyll-to-phytoplankton carbon varies by at least an order of
magnitude due to physiological responses of phytoplankton to their ambient environment (e.g. nutrients,
light, and temperature) (Cullen, 2015; Fennel and Boss, 2003; Geider, 1987). Thus, changes in
chlorophyll may result from physiologically induced modifications of the chlorophyll-to-phytoplankton
ratio rather than actual changes of phytoplankton biomass (Fommervault et al., 2017; Mignot et al., 2014).
Satellite surface chlorophyll alone is therefore likely insufficient for model validation and for constraining
biogeochemical models via parameter optimization.

80         Recent advances in autonomous platforms and sensors have opened opportunities for simultaneous





measurement of several biological and chemical properties throughout the upper ocean with high
resolution, over broad spatial scales and for sustained periods (Roemmich et al., 2019). In particular, the
biogeochemical (BGC) Argo program (Johnson and Claustre, 2016; Roemmich et al., 2019) will provide
temporally evolving 3D information on biogeochemical variability at previously unobserved scales. Here
we assess to what degree observations of chlorophyll fluorescence and particle backscatter from Argo
profiles improve the prospects of optimizing a biogeochemical model for the Gulf of Mexico.
Since the high computational cost and storage demands of 3D models make direct application of
most parameter optimization techniques difficult (but see Mattern et al., 2012; Mattern and Edwards,
2017; Tjiputra et al., 2007 for exceptions), they are typically applied in computationally efficient 1D
models before using the resulting parameters in 3D version (e.g. Hoshiba et al., 2018; Kane et al., 2011;
Kuhn and Fennel, 2019; Schartau and Oschlies, 2003). We follow the latter approach here.
The main objective of this study is to assess the added value of bio-optical profile information from
Argo floats for biogeochemical model optimization in the Gulf of Mexico. We first examine the feasibility
of improving the 3D model by applying the optimal parameters from 1D model optimizations. We find
that the gains from the 1D optimizations transfer to the 3D version. Then, by using different combinations
of satellite and float observations we show that parameters optimized with respect to satellite data cannot
reproduce subsurface distributions unless the float observations (i.e. chlorophyll, phytoplankton, and POC)
are also used.

## 2.  Study region

The Gulf of Mexico (GOM) is a semi-enclosed marginal sea (Figure 1) which is characterized by
eutrophic coastal waters on the northern shelf and an oligotrophic deep ocean. The high productivity in
the northern coastal region is fueled by large nutrient and freshwater inputs from the Mississippi and
Atchafalaya Rivers. The large nutrient load and strong stratification driven by Mississippi and
Atchafalaya River inputs lead to summer hypoxia and ocean acidification in bottom waters on the
northern shelf (Laurent et al., 2017; Yu et al., 2015), but nutrient export across the shelf break into the
open Gulf is minor (Xue et al., 2013).



The deep ocean of the GOM is oligotrophic. Previous satellite-based studies have revealed a clear
seasonal cycle in surface chlorophyll with highest concentrations in winter and lowest in summer
(Martínez-López and Zavala-Hidalgo, 2009; Muller-Karger et al., 1991, 2015). Thanks to advances in
autonomous profiling technology, recent studies based on simultaneous measurements of subsurface
chlorophyll and backscatter have demonstrated that the seasonal variability of surface chlorophyll might
be a result of the vertical redistribution of subsurface chlorophyll and/or physiological response to solar
radiation of phytoplankton (Fommervault et al., 2017; Green et al., 2014).
**3.    Methods**
**3.1. Biological observations**
Satellite-derived chlorophyll from the Ocean-Colour Climate Change Initiative project (OC-CCI,
https://www.oceancolour.org) with a spatial resolution of 4 km from 2010 to 2015 is used for model
validation and parameter optimization. These data were provided by the European Space Agency (ESA),
which produced a set of validated and error-characterised global ocean-color products by merging
SeaWiFS (Sea-viewing Wide Field-of-view Sensor), MODIS (Moderate-resolution Imaging
Spectroradiometer), and MERIS (medium-spectral resolution imaging spectrometer) products.
In addition to the satellite-based measurements, bio-optical measurements from six autonomous
profiling floats are used (Figure 1), which were deployed by the Bureau of Ocean Energy Management
(BOEM) and operated in the deep GOM from 2011 to 2015. These floats were equipped with a CTD and
bio-optical sensors to collect biweekly profiles of temperature, salinity, chlorophyll fluorescence, and
backscatter at 700 nm ($bbp700$ (m$^{-1}$)) from the surface to 1000 m depth (see Fommervault et al., 2017
and Green et al., 2014 for more details). Chlorophyll was derived from fluorescence based on the sensor
manufacturer's calibrations and cross-validated with the satellite estimates of surface chlorophyll. While
the surface chlorophyll measurements from the floats and the satellite estimates both show a typical
seasonal cycle and are highly correlated ($R^2$=0.74; see Figures S1 and S2a in the Supplement), the satellite
underestimates the float-measured chlorophyll concentrations in winter (Figure S1c). Satellite estimates
were therefore corrected following the regression equation shown in Figure S2a (Figure S1c).
The backscatter sensor carried by the floats provides the volume scattering function at a centroid





angle of 140° and a wave length of 700 nm ($\beta(140^o, 700nm)$  m$^{-1}$ sr$^{-1}$) (Green et al., 2014). The profiles
were filtered following Briggs et al. (2011) to remove spikes. To cross-validate the float-measured $bbp700$
with the satellite estimates, the $bbp670$ from OC-CCI was firstly converted to $bbp700$ through a power
law (Boss and Haëntjens, 2016):

$$bbp(\lambda 1) = \left(\frac{\lambda 1}{\lambda 2}\right)^{-\gamma} bbp(\lambda 2), \qquad (1)$$

where $\lambda 1$  and  $\lambda 2$  represent the measured wavelength, and  $\gamma$  was estimated as 0.78 based on the global
measurements. Compared to surface chlorophyll, surface $bbp700$ has a less distinct seasonal cycle (Figure
S3). For example, the coefficient of variation, defined as the ratio between standard deviation and mean
to show the extend of variability, is much lower for $bbp700$ (0.09 and 0.07 for floats and satellite,
respectively) than for chlorophyll (0.31 and 0.26 for floats and satellite, respectively). The $bbp700$ from
the floats is weakly correlated with the satellite estimates ($R^2$=0.11) and generally lower by a factor of
~0.45 than the satellite estimates (Figure S2b). The bbp700 profiles were therefore multiplied by 2.2
before being converted to $bbp470$ following the equ. 1.
Profiles of phytoplankton and POC were derived from the validated $bbp470$ profiles based on the
following empirical relationships

$$Phy = 30100 \times (bbp470 - 76 \times 10^{-5})\frac{1}{12 \times 6.625}, \qquad (2)$$


$$log10(POC) = 1.22 \times log10(bbp470) + 5.15. \qquad (3)$$

The relationships for phytoplankton (Martinez-Vicente et al., 2013; equ. 2) and POC (Rasse et al., 2017;
equ. 3) were obtained from a data set for the Atlantic Ocean that covers a wide range of oceanographic
regimes from eutrophic to oligotrophic ecosystems. The scale factors of 12 and 6.625 in equ. 2 represent





the molecular weight of carbon and the Redfield ratio to convert phytoplankton concentrations from mg
C m$^{-3}$ to mmol N m$^{-3}$. According to Behrenfeld et al. (2005), the intercept $76 \times 10^{-5}$ in equ. 2 represents
the background backscatter of nonalgal detritus. In this study, both chlorophyll and phytoplankton
approach zero when *bbp470* is $76 \times 10^{-5}$ m$^{-1}$, implying that the use of equ. 2 is appropriate.

## 63 3.2. 3D model description

The physical model is configured based on Regional Ocean Modeling System (Haidvogel et al.,
2008; ROMS, https://www.myroms.org) for the Gulf of Mexico (Figure 1). The model has a horizontal
resolution of 6~7 km and 36 terrain-following sigma layers with refined resolution near the surface and
bottom. The model solves for the horizontal and vertical advection of tracers using the Multidimensional
positive definitive advection transport algorithm (MPDATA, Smolarkiewicz and Margolin 1998).
Horizontal viscosity and diffusivity are parameterized by a Smagorinsky-type formula (Smagorinsky,
1963), and vertical turbulent mixing is calculated by the Mellor-Yamada 2.5-level closure scheme (Mellor
and Yamada, 1982). Bottom friction is specified by a logarithmic drag formulation with a bottom
roughness of 0.02 m. The model is forced by 3-hourly surface heat and freshwater fluxes, 6-hourly air
temperature, sea level pressure and relative humidity, and 10-m winds from the European Centre for
Medium-Range Weather Forecast ERA-Interim product with a horizontal resolution of 0.125$^o$ (ECMWF
reanalysis, https://www.ecmwf.int/en/forecasts/datasets/reanalysis-datasets/era-interim). A bulk
parameterization is applied to calculate the surface net heat fluxes and wind stress. The model is one-way
nested inside the 1/12$^o$ data-assimilative global HYCOM/NCODA (https://www.hycom.org). Tidal
constitutes are neglected in the model.
The biogeochemical model uses a 7-component model (Fennel et al., 2006) to simulate the nitrogen
cycle in the water column. The model describes the dynamics of two species of dissolved inorganic
nitrogen (nitrate, NO3, and ammonium, NH4), one function of phytoplankton (Phy), chlorophyll (Chl) as
a separate state variable which allows photo-acclimation based on the model of (Geider et al., 1997), one
function of zooplankton (Zoo), and two pools of detritus (i.e. small suspended detritus, SDeN, and large
fast-sinking detritus, LDeN). Water-sediment interactions are simplified by an instantaneous
remineralization parameterization, where detritus sinking out of water column immediately results in a





corresponding influx of ammonium into the bottom layer. Detailed descriptions of the model equations
can be found in Fennel et al. (2006). The biological model parameters are listed in Table 1.
The model receives freshwater, nutrients (NO3 and NH4) and organic matter inputs from major rivers
along the Gulf coast. Freshwater and nutrients from the Mississippi and Atchafalaya rivers are prescribed
based on the daily measurements by the US Geological Survey river gauges. River particulate organic
nitrogen (PON) is assigned to the small detritus pool and determined as the difference between total
Kjeldahl nitrogen and ammonium (Fennel et al., 2011). Other rivers utilize the climatological estimates
of freshwater, nutrients, and PON as in Xue et al. (2013).
Initial and open boundary conditions for NO3 are specified by applying an empirical relationship
between NO3 and temperature, derived from the World Ocean Atlas (WOA; Figure S4a), that is applied
to the temperature fields from HYCOM/NCODA. Analogously, empirical relationships between
chlorophyll and density (Figure S4b), phytoplankton and density (Figure S4c), and POC and density
(Figure S4d) were obtained from the median profiles of the bio-optical floats and used to derive initial
and boundary conditions for these variables. Zooplankton and small detritus were assumed to amount to
10% of phytoplankton biomass and the remaining fractions of POC attributed to large detritus.
A 6-year (5 January 2010 – 31 December 2015) hindcast was performed that includes the period of
operation of the bio-optical floats. The first year is considered model spin-up and the next five years will
be discussed.
**3.3. 1D model description**
As optimizing a 3D biogeochemical model is computationally expensive, it is more practical to
perform the optimization using a reduced-order model surrogate. A surrogate can be a coarser resolution
model, a simplified model, or a reduced-dimension model. In this study, a 1D model is used to optimize
the biological parameters of the 3D model. This approach has been successfully used previously (Hoshiba
et al., 2018; Kane et al., 2011; Oschlies and Schartau, 2005).
The 1D model, which is similar to that used by Lagman et al. (2014) and Kuhn et al. (2015), covers
the upper 200 m of the ocean with a vertical resolution of 5 m and is configured at one location in the
central Gulf (see Figure 1). In the vertical direction, the water column is divided into two layers: the





turbulent surface layer and a quiescent layer below. A higher diffusion coefficient ($K_{Z1} =$
$\max(H_{MLD}^2/400,10)$) is applied in the turbulent surface layer and a lower diffusion coefficient ($K_{Z2} =$
$K_{Z1}/2$) is assigned to the quiescent bottom layer. The interface between these two layers is determined
by the mixed layer depth ($H_{MLD}$), defined as the depth where the temperature is 5°C lower than at the
surface, and was obtained from a simulation of the 3D model. The model is integrated in time using the
Crank-Nicolson scheme for vertical turbulent mixing and an implicit time-stepping scheme for the
biogeochemical tracers, which are treated identically to the 3D model. Some of the biogeochemical
parameterizations require input of temperature and solar radiation, which are also taken from the 3D
model. As the 1D model does not consider horizontal and vertical advection, NO3 below 100 m is nudged
to that from the 3D base simulation with a nudging time scale of 20 days. The model is run for the year
2010 repeatedly for three cycles, with the first two are model spin-up and the last annual cycle used to
calculate the misfit between model and observations.
**3.4. Parameter optimization method**

The evolutionary algorithm described by Kuhn et al. (2015, 2018) is used to search for optimal

model parameters by minimizing the misfit between model and observations. The misfit is measured by
the following cost function:

$$F(\vec{p}) = \sum_{v=1}^{V} F_v(\vec{p}), \tag{4}$$

$$F_v(\vec{p}) = \frac{1}{N_v \sigma_v^2} \sum_{i=1}^{N_v} \left(\hat{y}_{i,v} - y_{i,v}(\vec{p})\right)^2, \tag{5}$$

where $\vec{p}$ represents the parameters vector, $V$ is the number of different observation types, $N_v$ is the
number of observations for each variable, $F_v(\vec{p})$ is the misfit for observation type $v$ measured as the
mean-square difference between observations ($\hat{y}$) and corresponding model estimates ($y(\vec{p})$). The cost



function $F_v(\vec{p})$ is normalized by the standard deviation of each variable type ($\sigma_v$) in order to remove the
effect of different units.
The algorithm is inspired by the rules of natural selection. Following Kuhn et al. (2015), an initial
parameter population of 30 parameter vectors is randomly generated within a predefined range of
parameters (see Table 1). The model is evaluated for each parameter vector and the resulting cost function
is calculated. For this initial generation and each of the following generations, the half of the population
with the lower misfit survives into the next generation. The other half is regenerated through a
recombination of survivors in a process analogous to genetic crossover. In addition, each newly generated
population is subject to random mutations by multiplying the parameter values by a random value
between 0 and 2. Parameter values exceeding the predefined range are replaced by their corresponding
minimum or maximum limits to avoid unrealistic values. The above procedure is performed iteratively
for 300 generations to reach the minimum of the cost function, which corresponds to the optimal
parameter set.
Previous parameter optimization studies have shown that it is difficult to constrain all model
parameters even for very simple ecosystem models because the information content of available
observations is typically insufficient (Matear, 1995; Fennel et al., 2001; Ward et al., 2010). Here we
conducted sensitivity tests to identify the parameters that are most sensitive to the available observations
and chose a subset of these to be optimized. In the **base case**, all parameters were at their initial guess
values obtained from the previous literature and some initial tuning. Then the **test cases** were run multiple
times by incrementally changing one parameter at a time to be the minimum, the first, second and third
quartile, and the maximum of its corresponding range while setting the other parameters to their initial
guess value (Table 1). The sensitivity was measured as the sum of a normalized absolute difference
between the base case ($y_{Base}$) and the test case ($y_{Test}$)

$$Q(y, \vec{p}) = \frac{1}{m} \sum_{i=1}^{m} \frac{1}{n} \sum_{j=1}^{n} \frac{|y_{Base} - y_{Test}|}{y_{Base}} \tag{6}$$






where *m* is the number of parameter increments (here 5) and *n* is the number of base-test pairs including
all 1D model grid cells throughout the whole simulation period for all variables to be compared.
Results of the sensitivity analysis are shown in Figure 2, where parameters are ranked by sensitivity
with respect to chlorophyll (Figure 2a) and the sum of chlorophyll, phytoplankton, and POC (Figure 2b).
POC is the sum of phytoplankton, zooplankton, and small and large detritus.

### 3.5. Parameter optimization experiments

For the parameter optimization of the 1D model, satellite chlorophyll within a 3×3 pixel (12 km×12
km area) around the 1D station and climatological monthly averages of the profiles from the bio-optical
floats were used.
To assess the effects of the optimization with respect to the different observation types, we conducted
three groups of experiments in which (A) surface satellite chlorophyll only, (B) surface satellite
chlorophyll and float profiles of chlorophyll, and (C) surface satellite chlorophyll and float profiles of
chlorophyll, phytoplankton, and POC were used. For each of these three groups, four to five optimizations
were conducted starting with the three most sensitive parameters and then adding one more parameter at
a time (Table 2) guided by the sensitivity analysis with respect to observed variables they used.
Specifically, groups A and B were based on the sensitivity analysis with respect to chlorophyll, while
group C was based on sensitivity analysis with respect to the sum of chlorophyll, phytoplankton, and
POC. Each optimization was replicated four times. The optimization with smallest model-data misfit
within each group was then used. Prior tests have shown that the available observations cannot
simultaneously constrain the sinking rates of small and large detritus ($w_{SDet}$ and $w_{LDet}$). Therefore, a
constant ratio of 0.1 between these two parameters ($w_{SDet} = 0.1 \times w_{LDet}$) was imposed and only one of the
two was optimized. In groups A and B, the aggregation parameter $\tau$ was fixed at 0.05 because prior tests
generated unreasonably high values for this parameter.
We report two different metrics of misfits for these groups of experiments. The first metric, which
we refer to as the case-specific cost function value, is based on the optimized observations in a given
experiment and is minimized by the optimization algorithm, i.e.



$$F_A(\vec{p}) = F_{SurfCHL}(\vec{p}), \tag{8}$$

$$F_B(\vec{p}) = F_{SurfCHL}(\vec{p}) + F_{CHL}(\vec{p}), and \tag{9}$$

$$F_C(\vec{p}) = F_{SurfCHL}(\vec{p}) + F_{CHL}(\vec{p}) + F_{Phy}(\vec{p}) + F_{POC}(\vec{p}). \tag{10}$$

However, the models with lower case-specific misfit does not necessarily have better predictive skill in reproducing the unoptimized observations because of the so-called overfitting problem, e.g. the model may be tuned to reproduce optimized observations through wrong mechanisms (Friedrichs et al., 2006). To account for this, a second metric referred to as the total misfit is given by equ. 10. For group C, the second metric is the same as the case-specific cost function. For groups A and B, the total misfit metric allows us to assess improvements in the model's predictive skill to represent unoptimized fields.

## 4.  Optimization of 1D models

### 4.1. Observations and base case

To provide context for the evaluation of our optimization experiments, the observations and the base case will be described first. As shown in Figure 3a, the observed surface chlorophyll shows a clear seasonality with the high concentrations in winter and low concentrations in summer. In the base case, the simulated surface chlorophyll fits observations well. Unlike the surface chlorophyll, the vertically integrated chlorophyll as well as the phytoplankton and POC over the upper 200 m tend to be more constant with much less seasonality (Figure 3b-d). This has been reported by Fommervault et al. (2017) who attributed the seasonality of surface chlorophyll to the vertical redistributions of subsurface chlorophyll and/or photoacclimation, rather than real changes in biomass.

The DCM is a ubiquitous phenomenon in the oligotrophic regions (Cullen, 2015). Observations detect a predominant DCM at around 60-100m depth throughout the whole year, with a sharp deepening in June and gradual shoaling after July (Figure 3e), reflecting the seasonality of the solar radiation. Unlike the large variability in the depth of the DCM, its magnitude is relatively constant at around 0.62 mg m$^{-3}$ (Figure 3f). In the annually averaged profiles, the observed DCM is located at about 80 m depth with a





concentration of 0.52 mg m$^{-3}$ (Figure 4a). The base case succeeds in reproducing the DCM at 65±7m
depth. However, it fails to reproduce the deepening of the DCM in June and the simulated annually
averaged depth of DCM is shallower by about 15 m than the observed. The simulated magnitude of the
DCM is about 2-fold larger than the observed (Figure 3f and Figure 4a) and hence the base case generally
overestimates vertically integrated chlorophyll (Figure 3b).

22        With respect to phytoplankton and POC, the observed maximum concentration occurs at about 60

m depth, which is 20 m above the DCM (Figure 4b-c). The observed vertical distributions of
phytoplankton and POC are not well captured by the base case. For example, phytoplankton and POC in
the upper layer are overestimated with the model-data discrepancies exceeding the variability of the
observations (Figure 4b-c). As a result, the base case yields an overall overestimation of the vertically
integrated phytoplankton and POC (Figure 3c-d).

Figure 4b also shows that both observed and simulated phytoplankton approach zero at about 160

m depth. Unlike phytoplankton, the observations show that the POC concentrations are 19 mg C m$^{-3}$ at
about 200 m depth because of the existence of detritus (Figure 4c). However, the base case fails to
reproduce this non-zero POC concentrations, indicating that the model might be underestimating the
carbon export fluxes at 200 m.
**4.2. Results of the optimizations**
**4.2.1.  Model-data misfits**

The case-specific cost function values and total misfits for the different 1D optimizations are shown

in Figure 5. Not surprisingly, all optimizations result in a reduction of the case-specific cost function
values. The extent of the reductions depends on the specific subset of parameters that were optimized.
However, the total misfits are not reduced in all optimizations. Optimizations A1 and A2 lead to slightly
larger total misfits than the base case and optimization B2 leads to a significantly larger total misfit. The
smallest total cost function values are achieved in A4, B4, and C4, i.e. in the experiments where a larger
subset of parameters was optimized (6 parameters). The optimal parameter sets (A4, B2, and C4), which
are selected based on case-specific misfit from these three groups, will be used in subsequent analyses





and hereafter are denoted simply as experiment A, experiment B, and experiment C. Further comparisons
are presented below to assess the impact of the different combinations of observations.

### 4.2.2. Experiment A

The optimal parameters from experiment A yield a 58% reduction in the misfit for surface
chlorophyll (Figure 5d). However, the vertical structure of chlorophyll deteriorates relative to the base
case (Figure 4a) because of inappropriate estimates of the initial slope ($\alpha$=0.0101; see table 2) and the
maximum ratio of chlorophyll to carbon ($\theta_{max}$=0.0191; see table 2). The annually averaged depth of the
DCM is lifted up to around 30±10m and the magnitude of DCM strongly decreases (Figure 3a, 4b).
Similar to chlorophyll, these deteriorations also manifest in the vertical phytoplankton and POC
distributions (Figure 4b-c). As a result, experiment A underestimates vertically integrated chlorophyll,
phytoplankton, and POC (Fig. 3b-d).

### 4.2.3. Experiment B

Due to the addition of observed chlorophyll profiles to the optimization in experiment B, the misfits
for surface and subsurface chlorophyll decrease relative to the base case (Figure 5d), but the reduction in
the misfit for surface chlorophyll (38%) is smaller than that in experiment A (58%). A straightforward
interpretation is that the addition of subsurface observations reduces the model's degrees of freedom to
fit one single observation type (surface chlorophyll). The vertical profile of chlorophyll is reproduced
better in experiment B than in the base case and experiment A in that the magnitude of the DCM is closer
to the observations, although the DCM depth is still too shallow, on average by about 20 m (Figure 4a).
The improvement in the vertical chlorophyll structure results in a better model-data fit of vertically
integrated chlorophyll (Figure 3b).
Despite the improvements in chlorophyll, the vertical profiles of phytoplankton and POC exhibit a
marked deterioration relative to the base case and experiment A (Figure 4b-c) because the parameter
optimization underestimates the maximum chlorophyll-to-carbon ratio ($\theta_{max}$ =0.0158; see table 2).
Experiment B leads to an overestimation of phytoplankton and POC relative to the base case with misfits
roughly 2.7 and 1.6 times larger than those of the base case, respectively (Figure 5d). Although
experiment B reproduces the non-zero POC concentrations at about 200 m depth, the proportion of



phytoplankton in the POC pool is incorrect. In contrast to the observations where detritus dominates POC,
the simulated POC at 200 m is dominated by phytoplankton (49%) followed by zooplankton (39%).

### 4.2.4.  Experiment C

Incorporating all observations (i.e. surface chlorophyll and profiles of chlorophyll, phytoplankton,
and POC) in experiment C improves the model-data misfits for almost all aspects except for surface
chlorophyll (Figure 3). Although a slight increase in the misfit occurs for the surface chlorophyll (~5%),
the total misfit is reduced by 75% compared to the base case. As shown in Figure 4a, the annually
averaged depth of DCM of 80 m coincides with the observed DCM, primarily because experiment C
reproduces the deepening of the DCM in summer. The magnitude of the DCM is also decreased relative
to the base case but remains higher than the observed. Phytoplankton and POC profiles exhibit only minor
deviations from the observations (Figure 4b-c). Importantly, experiment C reproduces the non-zero POC
concentrations at 200 m. In contrast to experiment B, phytoplankton in experiment C drops to zero at
about 160 m and POC is dominated by detritus (85%), which is more consistent with the observations.

### 4.3. Simulated carbon fluxes

Annually averaged carbon fluxes within the upper 200 m are shown for each experiment in Figure
6. The primary production in the base case amounts to 0.78 g C m$^{-2}$ day$^{-1}$, of which 37% is consumed by
zooplankton, and the remaining 63% flows into detritus pools through sloppy feeding, mortality, and
aggregation of phytoplankton. As for the production of detritus, contributions from the phytoplankton
and zooplankton pools account for 70% and 30%, respectively. Most of the produced detritus is recycled
into the nutrient pool fueling recycled primary production, and only a small fraction is removed from the
upper layer through gravitational sinking. As a result, carbon export, which is estimated as the sum of
sinking fluxes by phytoplankton and detritus, is only 0.00032 g C m$^{-2}$ day$^{-1}$ and accounts for 0.04% of
primary production.
Due to the underestimation of phytoplankton in experiment A, primary production is reduced to 0.21
g C m$^{-2}$ day$^{-1}$ in that case. All other fluxes in the top 200 m decrease relative to the base case as well,
except for the export flux which increases to about 0.8% of primary production. This relative increase in
export is the result of larger sinking rates of phytoplankton and detritus ($w_{Phy}$=0.95, $w_{LDet}$ =4.97; see table



2) than those used in the base case.

98       In contrast to experiment A, experiment B yields an increase of primary production relative to the

base case. The proportion of the grazing flux to primary production and the contribution of zooplankton
to the production of detritus also increase to about 59% and 52%, respectively. Unlike in the other three
experiments, carbon export in experiment B is dominated by the sinking of phytoplankton, reflecting its
large contribution to POC at 200 m. Although the simulated POC concentration at 200 m is very close to
the observations, the relative contributions of phytoplankton, zooplankton, and detritus are problematic
and likely do not result in a better estimation of carbon export (in this case 0.3% of primary production).

05       In experiment C, primary production is 0.30 g C m$^{-2}$ day$^{-1}$ with 24% flowing to zooplankton. The

mortality of zooplankton causes a flux of 0.047 g C m$^{-2}$ day$^{-1}$ to detritus, which accounts for 17% of the
production of detritus. Finally, about 24% of primary production is removed from the upper 200 m
through gravitational sinking. The simulated export ratio of 24% is within the wide range of reported
export ratios, from 6% to 43%, at 120 m depth in the Gulf of Mexico (see Table 3 of Hung et al., 2010).
Despite the high degree of uncertainty that exists when estimating export ratios (e.g., the global mean
export ratio varies from ~10% (Henson et al., 2012; Lima et al., 2014; Siegel et al., 2014) to ~20%
(Henson et al., 2015; Laws et al., 2000)), it is obvious that only experiment C reproduced an export ratio
of a reasonable magnitude. A more detailed validation of primary production and export fluxes will be
presented in the following sections.
**5.   3D biogeochemical model**

16       The optimal parameter sets from the 1D optimizations of A, B, and C were applied in the 3D model

for the whole GOM for five years (2011-2015). The performance of the 3D model can be regarded as a
cross-validation of the parameters optimized in 1D at different times and locations. It is possible that the
optimization algorithm has modified parameters to compensate for biases in the 1D simulations, e.g. the
absence of horizontal and vertical advection or the simplification of vertical diffusion, that degrades the
3D model performance. Indeed, directly applying the optimal parameter sets from 1D version to the 3D
model yields lower model-data agreement than the 1D counterpart and the following modifications to the
optimized parameters were made manually to bring the model-data agreement of 3D model in better



alignment with that of 1D version: the half-saturation for NH$_4$ uptake ($k_{NH4}$) was decreased to 0.01 in
experiment B and C, and the aggregation parameter ($\tau$) was decreased to 0.05 in the experiment C.

## 5.1. Spatial patterns of surface chlorophyll

First, the annual climatological surface chlorophyll from satellite and model are compared from 2011
to 2015. The satellite estimates show high chlorophyll in the coastal regions and low chlorophyll in the
deep ocean (Figure 7a). This spatial pattern of surface chlorophyll is well reproduced in all simulations
except in the experiment A which even fails to reproduce the relatively high chlorophyll near the
Mississippi-Atchafalaya river systems because of the high sinking rate of phytoplankton ($w_{Phy}$=0.95; see
Table 2). The largest model-data differences occur in the coastal regions, where all simulations
underestimate the observed surface chlorophyll because parameter optimization is only performed at one
station located in the deep ocean without considering the coastal environments. Based on this and the fact
that the floats operated in the deep ocean (Figure 1), only the model results in the deep ocean (depth >
1000 m) will be considered in the following discussion.

## 5.2. Subsurface distributions

Figure 8 shows the seasonal cycles of surface chlorophyll as well as the vertically integrated
chlorophyll, phytoplankton, and POC within the deep ocean (depth>1000 m). Analogous to the 1D
models, chlorophyll, phytoplankton, and POC are integrated over the upper 200 m. Comparisons of
vertical profiles between observations and model results are given in Figure 9. In general, the main
features in the 3D models are very similar to those in 1D. Experiment A cannot constrain the vertical
profiles of chlorophyll because of the inappropriate estimation of initial slope ($\alpha$), experiment B
overestimates phytoplankton and its contribution to POC since the maximum ratio of chlorophyll to
carbon ($\theta_{max}$) is weakly constrained, and experiment C shows significant improvements in the model-data
agreement. However, there are some differences between the 1D and 3D models. For example, the base
case of the 1D model overestimates the magnitude while underestimating the depth of the observed DCM.
Experiment B and C best improve the magnitude and depth of DCM, respectively. In contrast, in the 3D
model the vertical profile of chlorophyll and the magnitude of the DCM in the base case are already very
close to the observations and neither of the optimizations yield further improvement. These differences



between the 1D and 3D models might be a result of different spatio-temporal scales between the two
model versions, or the simplifications of physical processes in the 1D model.

53         We have also compared the chlorophyll-to-carbon ratio, primary production, and carbon export

fluxes from 1D and 3D models with observations (Figure 10). The chlorophyll-to-carbon ratio is
estimated as the vertically integrated chlorophyll divided by the phytoplankton in the upper 200 m (Figure
10a). As an important indicator of phytoplankton physiological status (Geider, 1987), the observed
chlorophyll-to-carbon ratio varies considerably in response to the ambient environment. In general, the
ratios derived from the 3D models are lower than their corresponding 1D values, but the differences are
still within the range of variability. Without utilizing the observations of phytoplankton and POC,
experiments A and B in both 1D and 3D versions underestimate the chlorophyll-to-carbon ratio. In
experiment C, the simulated chlorophyll-to-carbon ratios from 1D and 3D are in good agreement with the
observations although the observed variability is underestimated.

63         For reference, satellite-based primary production (PP) is provided by two algorithms, the Vertically

Generalized Production Model (VGPM, Behrenfeld and Falkowski 1997) and the Carbon-based
Productivity Model (CbPM, Westberry et al. 2008). As shown in Figure 10b, satellite-based PP differs
depending on the algorithm applied. PP results from all 3D simulations are qualitatively similar to the 1D
simulations. Experiment C provides the best estimates of PP when compared to satellite-based estimates
from VGPM and CbPM, both in 1D and 3D.

69         The base case and experiments A and B yield carbon export fluxes smaller by one to two orders of

magnitude than experiment C. Thus, only experiment C from the 1D and 3D models are shown in Figure
10b in comparison to observations from sediment traps (see supplementary material). The carbon export
fluxes at 200 m from the 1D and 3D are similar in magnitude although the 1D model yields higher fluxes
and larger variability. Despite the scarcity of carbon export observations in the GOM, the model estimates
are within the range of observations down to ~1,600 m and capture the observed declining trend of carbon
export with depth.

76         In summary, all the results above demonstrate the feasibilities of using the locally optimized

parameters from the 1D model to improve the 3D simulation. In addition, by incorporating all available

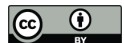



observations (i.e. surface chlorophyll from satellite estimates, profiles of chlorophyll, phytoplankton, and
POC from bio-optical floats), experiment C cannot only simulate the biogeochemical processes well in
the upper 200 m, but also reproduce the carbon export flux and its associated attenuation in the deep ocean
(200-1600m) of the GOM.
**6.   Discussion**
**6.1.   Trade-offs between different observations for parameter optimization**

84        The results of the optimization experiments vary dramatically depending on how many observation

types are used. Using only satellite surface chlorophyll in experiment A succeeds in reducing the misfits
of surface chlorophyll, but at the expense of the vertical structure since the predominant DCM disappears
in experiment A. Satellite surface chlorophyll alone cannot constrain several vital parameters, including
the initial slope of the productivity-irradiance curve ($\alpha$) and the maximum ratio of chlorophyll to carbon
($\theta_{max}$), with confidence. This result highlights the importance of subsurface observations for parameter
optimization and similarly for model validation.

91        The floats provide valuable subsurface observations, but chlorophyll profiles alone are not sufficient

for parameter optimization. In experiment B, the addition of chlorophyll profiles leads to significant
improvements in vertical chlorophyll distributions; however, the profiles of phytoplankton and POC
deteriorate largely because the maximum ratio of chlorophyll to carbon ($\theta_{max}$) is weakly constrained.
Using estimates of phytoplankton biomass and POC derived from backscatter measurements in
experiment C yields the best estimation of plankton-related state variables and carbon fluxes (i.e. primary
production and carbon export). Only in this experiment do the improvements obtained from observations
in the upper 200 m extend to the deep ocean as reflected in the improved carbon export estimates below
1,000 m.

00        It should be noted, however, that degradation of unoptimized variables did not occur in all

optimizations within experiments A and B. In some cases, the unoptimized fields were improved. For
example, the A2 optimization yields a 69% reduction in the misfit for subsurface chlorophyll (Figure 5d)
and large improvements of chlorophyll profiles (Figure S5a) even though no observations of subsurface
chlorophyll are used. Another example is that B1 optimization improves simulations of phytoplankton



and POC (Figure 5d and Figure S5b-c) through the correlations between the observed chlorophyll and
phytoplankton ($r^2 = 0.69$) and POC ($r^2 = 0.69$). Similar findings have been reported in Prunet et al. (1996b)
where the improvements of chlorophyll profiles within the mixed layer were obtained by using surface
chlorophyll in a 1D model. In a more recent study by Xiao and Friedrichs (2014a) where satellite data
was used subsurface fields were improved in addition to surface fields.
In optimizations A2 and B1, the improvement in unoptimized fields occurred because the poorly
constrained parameters were not optimized but well defined ($\alpha = 0.125$ in the optimization A2 and $\theta_{max}=$
0.0535 in the optimization B1; see table 2). Including the unconstrained parameters into the parameter
optimization can return a lower misfit with respect to the observations used in optimization but increases
the risk of overfitting and reduces the model's predictive skill, i.e. the ability to simulate unoptimized
observations and those collected at different locations and times. This is consistent with previous studies
(Friedrichs et al., 2006, 2007; Ward et al., 2010). For example, Friedrichs et al. (2006) optimized three
ecosystem models of different complexities against three seasons of observations and the resulting
parameters were used to quantify the predictive skill for the fourth season. Cross-validation showed that
exclusion of the poorly constrained parameters from the optimization increased the predictive skill.
Although prior knowledge of the parameters allows one to exclude those poorly constrained ones
from the optimization and thus can prevent degradation in unoptimized variables, most parameters are
poorly known. Thus, the ultimate resolution of this issue should be to increase availability of observations
so that more parameters can be constrained with confidence. In addition, even if the unconstrained
parameters are well-known, a lack of observations hampers our ability to recognize improvements in the
model's predictive skill and hence may prevent us from identifying the optimal solutions. For example,
without the observations of phytoplankton and POC, we could not have known that optimization B1
improved simulations of phytoplankton and POC, let alone that the optimization B1 was a better solution
than the optimization B2 (the experiment B) in terms of the lower total misfit as shown in Figure 5d.
It has been suggested that when performing a parameter optimization, not only parameter values but
also parameter uncertainties should be taken into account (Fennel et al., 2001; Ward et al., 2010;
Bagniewski et al., 2011). The parameter uncertainties can be assessed by performing an uncertainty





analysis (Fennel et al., 2001; Prunet et al., 1996a, 1996b), replicating the parameter optimization (Ward
et al., 2010), and cross-validating the resulting parameters (Xiao and Friedrichs, 2014a). In this study, a
cross-validation of the parameters was conducted by evaluating the model's predictive skill with respect
to different variables, times, and locations. Although this cross-validation at different times and locations
may give some indication of overfitting, it cannot determine whether the model reproduces observations
through wrong mechanisms because a small misfit of cross-validation can be caused by missing
validations of key variables or fluxes, e.g. ignorance of phytoplankton and PP in the experiment B, while
a large misfit can be a result of the intrinsic heterogeneity of biological parameters in different times
(Mattern et al., 2012) and locations (Kidston et al., 2011), e.g. underestimation of coastal surface
chlorophyll in the experiment C. Therefore, it is important to evaluate the predictive skill of unoptimized
variables.

43        Collectively, the discussion above highlights the values of BGC float data for parameter optimization

and model validation, not only because of their high spatio-temporal coverage but also their ability to
measure multiple properties simultaneously.

## 6.2. Feasibilities of applying the local optimized parameters to 3D models

47        As the high computational cost makes direct optimization for a 3D biogeochemical model

impractical, we performed parameter optimizations first in a 1D surrogate model with the same
biogeochemical component as the 3D model. However, there are some difficulties in porting the locally
optimized parameters to the basin-scale model. Firstly, the 1D model necessarily neglects horizontal
advection, which can result in differences between the 1D and 3D models. On the one hand, the optimized
parameters from the 1D model may have been adjusted to compensate for biases in the biological
properties caused by neglecting advection and, as a result, this may degrade the 3D simulations (Kane et
al., 2011). On the other hand, counter examples exist where the 3D simulations outperform the 1D models
(Hoshiba et al., 2018). Secondly, the spatial heterogeneity of parameters (e.g., Kuhn and Fennel 2019) is
another issue that influences the portability of parameters from 1D to 3D models. In some studies, the
parameter optimization has been performed at several contrasting stations with the goal of using different
parameter sets in different regions of the 3D model (Hoshiba et al., 2018). In other studies different





stations were optimized simultaneously to obtain one single optimized parameter set (Kane et al., 2011;
Oschlies and Schartau, 2005; Schartau and Oschlies, 2003). Such parameters compromise the misfit in
each single station but take account into all stations and can often yield an overall better simulation of all
these stations as shown by Kuhn and Fennel (2019).

63        In our study, the similarities in general features between the 1D and 3D models confirm the

portability of the resulting parameters in the deep ocean of the GOM while the underestimation of surface
chlorophyll in the coastal regions may result from the contrasting ecosystem functioning between coastal
regions and deep ocean. For example, the highly productive continental shelf in the northern GOM is
fueled by the large nutrient load from the Mississippi and Atchafalaya river systems with primary
production being as high as 4 g C m$^{-2}$ day$^{-1}$ near the Mississippi river delta (Fennel et al., 2011), while
the deep ocean is oligotrophic and nutrient limited with the primary production ranging from 0.2 to 0.5 g
C m$^{-2}$ day$^{-1}$ (see Figure 10).
7. **Conclusions**

72        In this study, we have performed parameter optimization for a 1D biogeochemical model and then

used the resulting parameters to generate simulations with a corresponding 3D model in the GOM. Three
experiments have been conducted by using different combinations of observations (surface chlorophyll
from satellite estimates, vertical profiles of chlorophyll, phytoplankton biomass and POC from BGC Argo
floats) in order to examine the trade-offs between the different observations for parameter optimization.
Two misfit metrics have been defined using the case-specific misfit and the total misfit to measure the
models' abilities to reproduce the optimized and unoptimized observations.

79        Model results show that satellite surface chlorophyll alone cannot reproduce well the vertical

structures in a biogeochemical model unless profile observations are used in addition. BGC Argo floats
are an excellent platform for obtaining such observations at high spatio-temporal coverage and for a
relatively broad suite of parameters. Only adding chlorophyll profiles is not sufficient because it fails to
constrain the ratio of chlorophyll to phytoplankton, but the addition of backscatter, which allows
estimation of phytoplankton biomass and POC, enables us to constrain the subsurface carbon state
variables and reproduce well PP and carbon export fluxes to below1000 m depth. Finally, our 3D model





was improved and reproduced similar results as the 1D version, which is promising for the application of
parameter optimization.

*Code and data availability:* The ROMS model code can be accessed at http://www.myroms.com (last
access: 16 June 2016). HYCOM data can be downloaded at http://tds.hycom.org/thredds/dodsC/datasets
(last access: 16 August 2018). Profiling data from the BGC-Argo floats are available at the National
Oceanographic Data Center (NOAA), https://data.nodc.noaa.gov/cgi-bin/iso?id=gov.noaa.nodc:159562
(Hamilton and Leidos, 2017)

*Author contributions.* BW and KF conceived the study. BW carried out optimization experiments, model
simulations and analyses. LY assisted with set-up and validation of the 3D model. CG assisted with
processing of the BGC float data. BW and KF discussed the results and wrote the paper with contributions
from the coauthors.

*Competing interests.* The authors declare that they have no conflict of interest.

*Financial support.* This research was funded by the Gulf of Mexico Research Initiative (GoMRI-V-487).









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





**Table list**
Table 1. Initial values and ranges of primary parameters used in the biogeochemical model

| Descriptions (unit) | Symbol | Value | Range |
|---|---|---|---|
| Radiation threshold for nitrification (W m$^{-2}$) | $I_0$ | 0.0095[a] | 0.005[b]-0.01[b] |
| Half-saturation radiation for nitrification (W m$^{-2}$) | $k_I$ | 0.1[a] | 0.01[b]-0.5[b] |
| Maximum nitrification rate (day$^{-1}$) | $n_{max}$ | 0.2[c] | 0.01[b]-0.35[b] |
| Phytoplankton growth at 0 ℃ (Dimensionless) | $\mu_0$ | 0.69[a] | 0.1[b]-3.0[b] |
| Initial slope of P-I curve (mg_C (mg_Chl W m$^{-2}$ day)$^{-1}$) | $\alpha$ | 0.125[a] | 0.007[a]-0.13[a] |
| Half-saturation for NO$_3$ uptake (mmol_N m$^{-3}$) | $k_{NO3}$ | 0.5[a] | 0.007[a]-1.5[a] |
| Half-saturation for NH$_4$ uptake (mmol_N m$^{-3}$) | $k_{NH4}$ | 0.5[a] | 0.007[a]-1.5[a] |
| Phytoplankton mortality (day$^{-1}$) | $m_P$ | 0.075 | 0.01[b]-0.2[b] |
| Aggregation parameter (day$^{-1}$) | $\tau$ | 0.1 | 0.01[b]-25[b] |
| Maximum chlorophyll to carbon ratio (mg_Chl mg_C$^{-1}$) | $\theta_{max}$ | 0.0535[c] | 0.005[a]-0.15[b] |
| Phytoplankton sinking velocity (m day$^{-1}$) | $w_{Phy}$ | 0.1[a] | 0.009[a]-25[a] |
| Maximum grazing rate (day$^{-1}$) | $g_{max}$ | 0.6[a] | 0.1[b]-4[b] |
| Half-saturation for phytoplankton ingestion ((mmol_N m$^{-3}$)$^2$) | $k_P$ | 0.5 | 0.01[b]-3.5[a] |
| Zooplankton assimilation efficiency (Dimensionless) | $\beta$ | 0.75[a] | 0.25[b]-0.75[b] |
| Zooplankton basal metabolism (day$^{-1}$) | $l_{BM}$ | 0.01 | 0.01[b]-0.15[b] |
| Zooplankton specific excretion (day$^{-1}$) | $l_E$ | 0.1[a] | 0.05[b]-0.35[b] |
| Zooplankton mortality (day$^{-1}$) | $m_Z$ | 0.2 | 0.02[b]-0.35[b] |
| Small detritus remineralization (day$^{-1}$) | $r_{SD}$ | 0.3[c] | 0.005[b]-0.25[a] |
| Large detritus remineralization (day$^{-1}$) | $r_{LD}$ | 0.1 | 0.005[b]-0.25[a] |
| Small detritus sinking velocity (m day$^{-1}$) | $w_{SDet}$ | 0.1[a] | 0.009[a]-25[a] |
| Large detritus sinking velocity (m day$^{-1}$) | $w_{LDet}$ | 1[a] | 0.009[a]-25[a] |

a Fennel et al. (2006); b Kuhn et al. (2018); c Yu et al. (2015)



Table 2. The best fit of parameter set for each experiment

| | $w_{Phy}$ | $m_P$ | $k_{NH4}$ | $\tau$ | $\theta_{max}$ | $\alpha$ | $w_{LDet}$ |
|---|---|---|---|---|---|---|---|
| **Base** | 0.1000 | 0.0750 | 0.5000 | 0.1000 | 0.0535 | 0.1250 | 1.000 |
| **A1** | 0.0608 | 0.0100 | 1.5000 | -- | -- | -- | -- |
| **A2** | 0.6863 | 0.0100 | 0.0195 | -- | 0.0169 | -- | -- |
| **A3** | 1.6567 | 0.1978 | 0.1004 | -- | 0.0250 | 0.0219 | -- |
| **A4** | 0.9468 | 0.0737 | 0.2454 | -- | 0.0191 | 0.0101 | 4.9694 |
| **B1** | 0.2863 | 0.0983 | 1.5000 | -- | -- | -- | -- |
| **B2** | 0.4217 | 0.0130 | 0.0300 | -- | 0.0158 | -- | -- |
| **B3** | 2.1016 | 0.0176 | 1.5000 | -- | 0.0346 | 0.0079 | -- |
| **B4** | 0.0090 | 0.0100 | 1.5000 | -- | 0.0361 | 0.0405 | 8.3514 |
| | $w_{Phy}$ | $r_{LD}$ | $m_P$ | $\tau$ | $k_{NH4}$ | $w_{LDet}$ | $\theta_{max}$ |
| **Base** | 0.1000 | 0.1000 | 0.0750 | 0.1000 | 0.5000 | 1.0000 | 0.0535 |
| **C1** | 1.9231 | 0.2500 | 0.1805 | -- | -- | -- | -- |
| **C2** | 0.9755 | 0.2500 | 0.0100 | 1.1402 | -- | -- | -- |
| **C3** | 0.4071 | 0.0630 | 0.0100 | 1.8531 | 0.0070 | -- | -- |
| **C4** | 0.0090 | 0.0050 | 0.0634 | 0.0995 | 0.0431 | 5.6623 | -- |
| **C5** | 0.0090 | 0.2245 | 0.0100 | 0.6451 | 1.5000 | 2.5202 | 0.0614 |



**Figure captions**

45

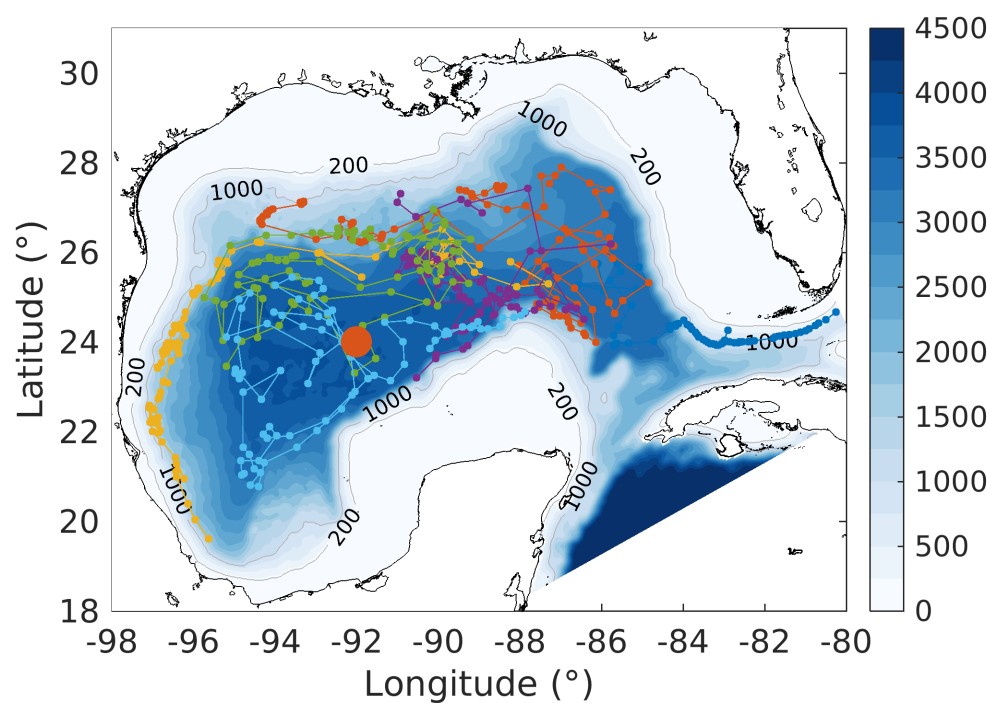

46

Figure 1. Model bathymetry (unit: m) with trajectories of six bio-optical floats (small colored dots and lines) which operated in the Gulf of Mexico from 2011 to 2015. The location of the 1D model is denoted by the large orange dot.





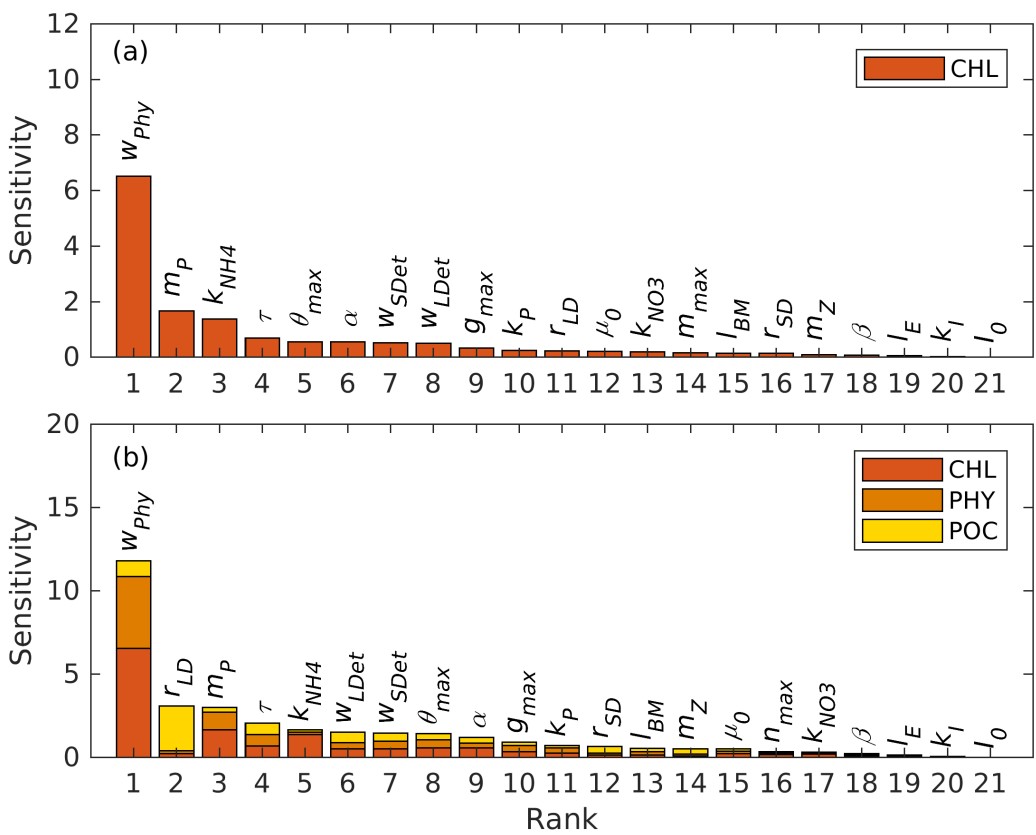

Figure 2. Parameter sensitivities (unit: dimensionless) with respect to (a) chlorophyll and (b) the sum of chlorophyll, phytoplankton, and POC.

Figure 3. Observed (black dots) and simulated (colored lines) annual cycle of surface chlorophyll (a), vertically integrated chlorophyll (b), vertically integrated phytoplankton (c), vertically integrated POC (d), and the depth (e) and magnitude (f) of the DCM. Chlorophyll, phytoplankton, zooplankton, and POC are integrated over the top 200 m.



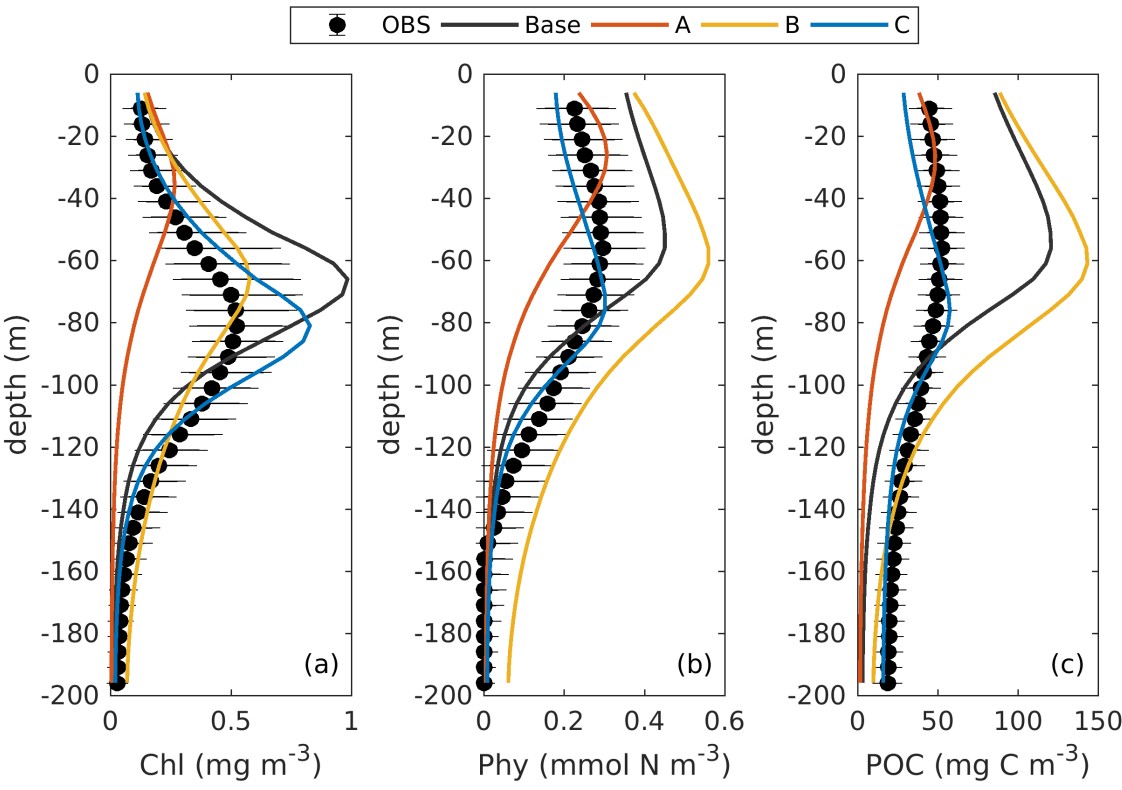

62

Figure 4. Observed (black dots with error bars) and simulated (colored lines) vertical profiles of

chlorophyll, phytoplankton, and POC.





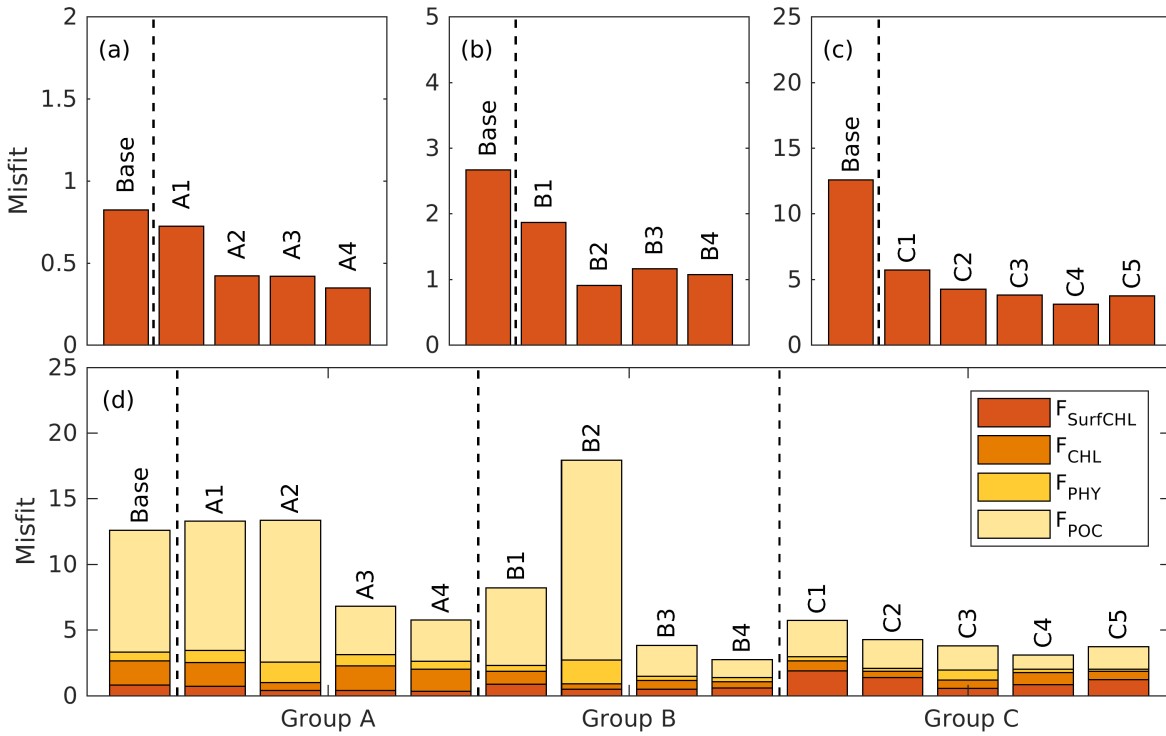

Figure 5. The case-specific cost function values (a-c) and total misfits (d) of the base case and the different optimizations.



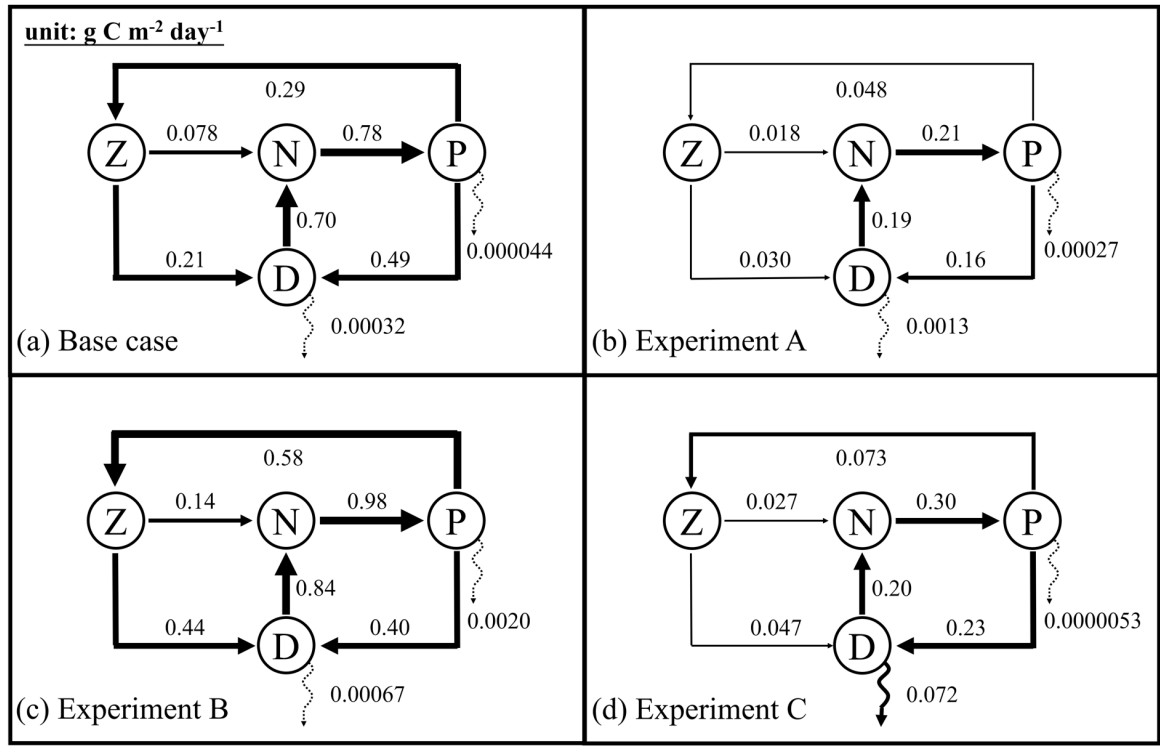

Figure 6. Annually averaged carbon fluxes integrated over the upper 200 m (unit: g C m$^{-2}$ day$^{-1}$) for the base case (a) and optimized experiments A, B, and C. The N, P, Z, and D stand for the pools of nutrient, phytoplankton, zooplankton, and the sum of small and large detritus, respectively. The thickness of arrows scales with the magnitude of fluxes. Dashed arrows represent fluxes lower than 0.01 g C m$^{-2}$ day$^{-1}$.


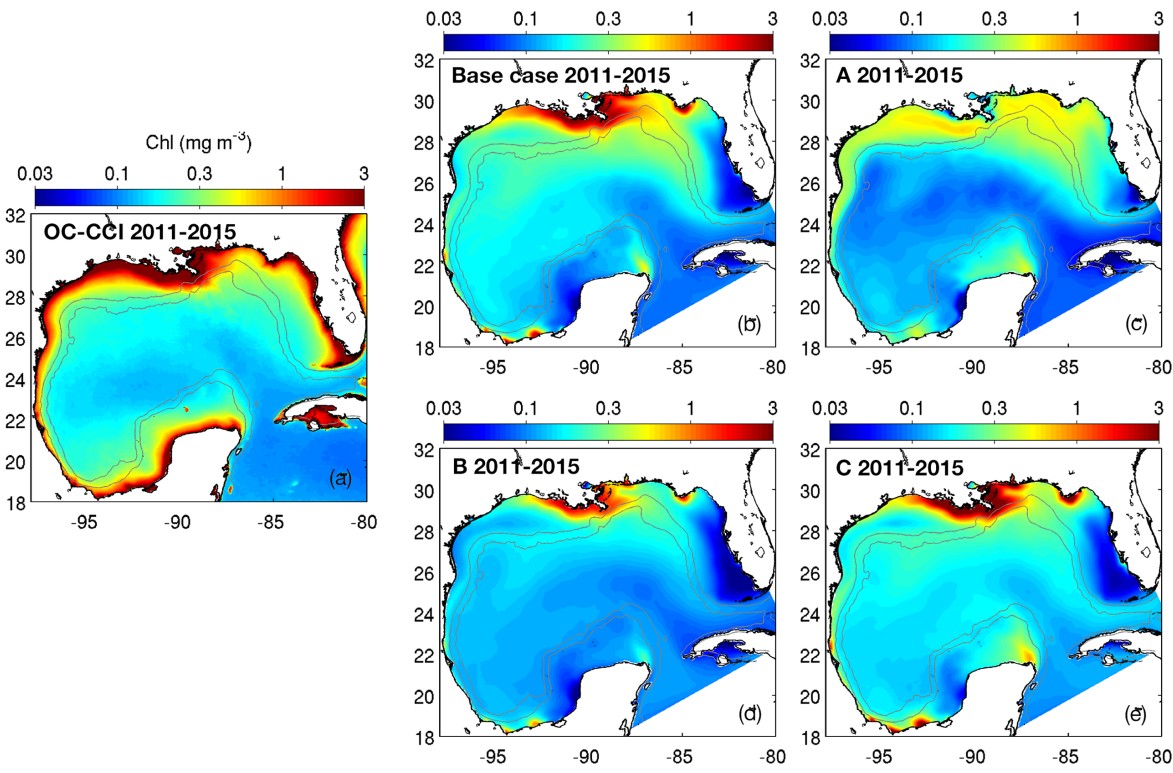

74

Figure 7. Spatial distributions of the annual mean chlorophyll in the surface layer from the satellite (OC-CCI) climatology (2011-2015) and the different model versions. The gray contours mark the bathymetric depths of 200 and 1000 m.

78

79

Figure 8. Observed and simulated seasonal cycles of surface chlorophyll (a), vertically integrated chlorophyll (b), vertically integrated phytoplankton (c), and vertically integrated POC (d) from each 3D models. Solid lines represent the median values over the deep ocean of GOM (depth>1000m). Error bars and shades show the 25% and 75% percentiles. Chlorophyll, phytoplankton, and POC are integrated over the top 200m.





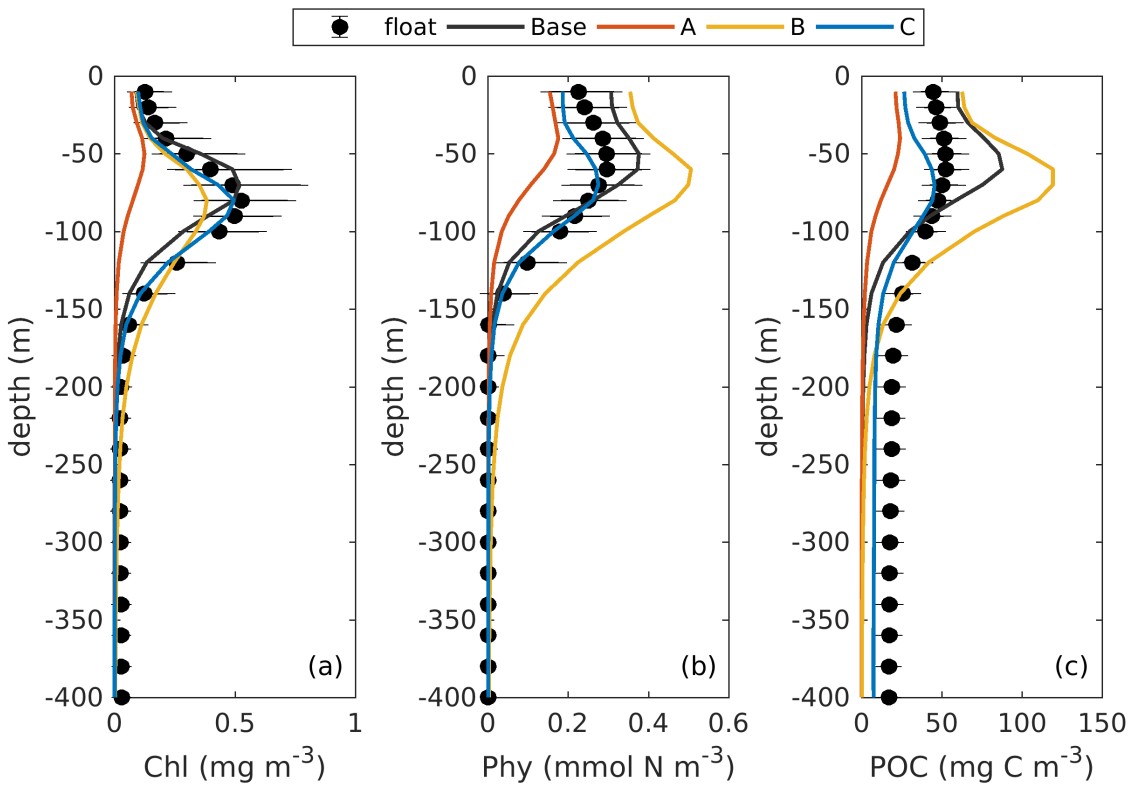

85

Figure 9. Observed and simulated vertical profiles of chlorophyll, phytoplankton, and POC from each

3D models.

88





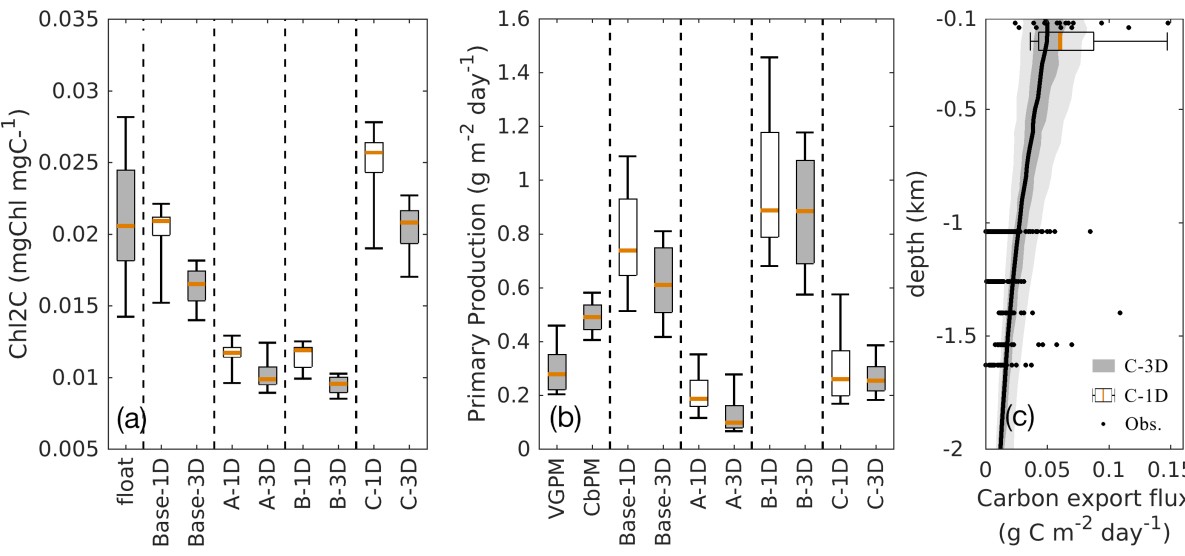

Figure 10. Comparisons of the chlorophyll to carbon ratio (a), primary production (b), and carbon export fluxes (c) between the 1D and 3D models.