# Peer review of "Assessing the value of BGC Argo profiles versus ocean colour observations for biogeochemical model optimization in the Gulf of Mexico"

_Biogeosciences, 2020_

## Referee Comment (RC1) · Anonymous Referee #1 · 21 May 2020

Review of the BG Discussion paper (**bg-2020-137**) "Assessing the value of BGC Argo profiles versus ocean colour observations for biogeochemical model optimization in the Gulf of Mexico" by Bin Wang et al.

Dear Editor, I read with great interest this manuscript related to the application of an optimization procedure combining satellite and BGC-Argo floats measurements with a biogeochemical model of the Gulf of Mexico. The manuscript is well organized and provides important insight to improve model parameterizations combining state-of the-art sensors data-streams. I found particularly important the results showing the importance of combining chlorophyll and bbp measured by BGC-Argo to optimize the model in terms of POC and export. In my opinion it is also important the results that, to a large extent, the optimization procedure carried out on the 1D model can be translated to the 3D model implementation.

Therefore, I suggest the publication of this manuscript after minor revisions reported below. My comments are indented in blue.

**Minor revisions 1. Introduction**

Pg2 line 45. because the number of parameters increases **exponentially** with the number of state variables (Denman, 2003).

I would expect a polynomial [quadratic] not an exponential increase of the parameter number vs state variable number. In fact, Denman (2003) provides a polynomial formula for the increase of fluxes vs state variables and elaborates the estimate of increase of parameters basing on this formula.

Pg3 lines 68-70. This is especially problematic in oligotrophic regions where the maximum chlorophyll concentration (referred as the deep chlorophyll maximum, DCM) is pronounced near the base of the euphotic zone because of photo-acclimation (Cullen, 2015; Fennel and Boss, 2003).

Could Authors explain better what they mean? In general DCM can appear in mathematical terms even without photo-acclimation (e.g. Varela et al. 1994, Ryabov and Blasius 2008). I suggest Authors to be more specific on this point. Nonetheless, the effect of photo-acclimation mentioned here could produce an enhancement of the DCM feature [already explainable by mechanisms other than photo-acclimation] and could be particularly important for the specific area of the GoM.

**References**

Varela, R. A., A. Cruzado, and Joaquín Tintoré. "A simulation analysis of various biological and physical factors influencing the deep-chlorophyll maximum structure in oligotrophic areas." *Journal of Marine Systems* 5.2 (1994): 143-157.

Ryabov, Alexei B., and Bernd Blasius. "Population growth and persistence in a heterogeneous environment: the role of diffusion and advection." *Mathematical Modelling of Natural Phenomena* 3.3 (2008): 42-86.

**2. Study Region**

I would suggest Authors to add information about the recirculation times of the GoM waters with

respect to the Atlantic Ocean boundary conditions (BC), this would give an idea on the relevance of the BC in the experiments.

Moreover it would be interesting to know what are the observed dominant plankton species, or plankton functional types, in the open-ocean part of GoM [pico-phytoplankton and flagellates?]. This could be useful to figure out the implications of the choice of considering one phytoplankton in the model when compared to data.

**3. Methods 3.1. Biological observations**

Pg 5 lines21-22 SeaWiFS (Sea-viewing Wide Field-of-view Sensor), MODIS (Moderate-resolution Imaging Spectroradiometer), and MERIS (medium-spectral resolution imaging spectrometer) products.

What about VIIRS? Is it included in the database?, please check.

Pg 5 line 26 and particulate backscattering

Pg 5 lines 31-32 Satellite estimates were therefore corrected following the regression equation shown in Figure S2a (Figure S1c).

Pg 6 lines 45-48 The bbp700 from the floats is weakly correlated with the satellite estimates (R2=0.11) and generally lower by a factor of ~0.45 than the satellite estimates (Figure S2b). The bbp700 profiles were therefore multiplied by 2.2 before being converted to bbp470 following the equ. 1.

In one case Authors consider the BGC-Argo the ground truth (Pg 5 lines 31-32) in the other Authors correct BGC-Argo with respect to satellite data (Pg 6 lines 45-48). The procedure seems a bit circular, could Authors explain better this part?

Pg 6 line 44 extent

**3.2. 3D Model description**

Pg 7 lines 74-75 Medium-Range Weather Forecast ERA-Interim product with a horizontal resolution of 0.125° (ECMWF reanalysis, https://www.ecmwf.int/en/forecasts/datasets/reanalysis-datasets/era-interim).

The resolution reported in the link above for ERA-Interim is 80 km this does not match with the  $0.125^{\circ}$  (~12 km) reported in the manuscript, could Authors double check?

**3.3. 1D Model description**

Pg8 line 10-12. The 1D model, which is similar to that used by Lagman et al. (2014) and Kuhn et al. (2015), covers the upper 200 m of the ocean with a vertical resolution of 5 m and is configured at one location in the central Gulf (see Figure 1).

Authors should add some comment about the choice of 5 m vertical resolution. What is the 3D model vertical resolution in the same region of the GoM? Why not taking 1m resolution that

probably is the vertical resolution of data acquired by BGC-Argo floats?

Pg9 line 13-15. A higher diffusion 213 coefficient ( $K_{ZI} = \max(H^2 \operatorname{MLD} / 400, 10)$ ) is applied in the turbulent surface layer and a lower diffusion coefficient ( $K_{Z2} = K_{ZI}/2$ ) is assigned to the quiescent bottom layer.

What is the unit of measure of  $K_{1,2}$ , please add this information in the text. If it is m2/s it seems very high, because even with  $H_{MLD} = 0 \rightarrow K_{ZI} = 10 \rightarrow K_{Z2} = 5$ . Please, add also unit of measure of  $H_{MLD}$ , meters?

**3.4. Parameter optimization method**

Pg 11 line 63. and n is the number of base-test pairs including

I suggest to use consisting rather than including that in my opinion is confusing. In fact n accounts exactly to what specified (i.e. base –test pairs etc etc )\_and not more.

The range of variability of Wphy spans many orders of magnitude and it is the most sensitive parameter, could Author comment on that?

**3.5. Parameter optimization experiments**

Pg 11 line 69-71. For the parameter optimization of the 1D model, satellite chlorophyll within a 3'3 pixel (12 km'12 km area) around the 1D station and climatological monthly averages of the profiles from the bio-optical floats were used.

Did you considered all the BGC-Argo data available or only the ones near the virtual mooring of your 1D experiment? How you decided the BGC-Argo float to include in the optimization procedure?

Pg 11 line 81-84. Prior tests have shown that the available observations cannot simultaneously constrain the sinking rates of small and large detritus (wsDet and wLDet)Therefore, a constant ratio of 0.1 between these two parameters (wsDet=0.1 wLDet) was imposed and only one of the two was optimized.

I suggest Authors to be clearer: I cannot get why the two parameters cannot be constrained. In general, after you completed your optimization procedure you could perturb the parameter 0.1 (ratio between WsDet and WlDet) to see if it corresponds to a minimum for your metrics F(p) or if there are better values other than 0.1. Or the system is unstable if you don't take 0.1? Please explain.

Pg 12 line 92 please remove and from the equation.

**4. Optimization of 1D models**

General question: how you define the DCM depth in the presented analyses?

**4.1. Observations and base case**

Pg 12. Lines 107-109. Unlike the surface chlorophyll, the vertically integrated chlorophyll as well as the phytoplankton and POC over the upper 200 m tend to be more constant with much less seasonality (Figure 3b-d).

This statement refers to observation? Please specify. Is it possible to add the error bar to the dots of Figure 3 as in the case of Figure 4?

Pg 12 line 14 in June and gradual shoaling after July (Figure 3e), reflecting the seasonality of the solar radiation.

With the term *reflecting* Authors mean that there is direct causality or correlation?

Pg 13 line 18 However, it fails to reproduce the deepening of the DCM in June

This deepening is related to a physical process (change in some environmental regulating factor) or to a biogeochemical process?

**4.2. Results of the optimizations**

**4.2.1 Model-data misfits**

**4.2.2 Experiment A**

Pg 13 line 41-42 The optimal parameter sets (A4, B2, and C4), which are selected based on case-specific misfit from these three groups, will be used in subsequent analyses ...

Why Authors decide to use B2 rather than B4 that show a smaller total misfit? Is it better to take a realization with better case-specific misfit or better total misfit?

**4.2.3 Experiment B**

Pg 14 line 47-49 However, the vertical structure of chlorophyll deteriorates relative to the base case (Figure 4a) because of inappropriate estimates of the initial slope ( $\alpha$ =0.0101; see table 2) and the maximum ratio of chlorophyll to carbon ( $\theta$ max=0.0191; see table 2).

If Authors can judge a-priori that the values for  $\alpha$  and  $\theta_{max}$  resulting from the optimization are not appropriate, why they didn't consider different parameter ranges in the optimization procedure from the beginning, excluding bad values?

Pg 14 lines 57-59. A straightforward interpretation is that the addition of subsurface observations reduces the model's degrees of freedom to fit one single observation type (surface chlorophyll).

Does this imply that a model with more parameters (e.g. more phytoplankton species) would fit better?

Pg15 line 70 In contrast to the observations where detritus dominates POC ...

What observation? Please add reference.

**4.2.4 Experiment C**

Pg 15 lines 76-78 As shown in Figure 4a, the annually averaged depth of DCM of 80 m coincides with the observed DCM, primarily because experiment reproduces the deepening of the DCM in summer.

Interesting. Can Authors explain if there is a specific parameter/mechanism that controls this dynamical deepening of the DCM? Or it is a complex combination of parameters values generating this emergent property?

**5.3D biogeochemical model**

Could Author explain better the manual correction described at pg 17 lines 24-25? If they consider the corrected values more realistic why they didn't narrowed the parameter variability range in the optimization experiment? Or the 1D vs 3D implementations do not allow this? For example, is there a simple explanation for the need to set  $K_{NH4}$  to 0.01 in the 3D experiments? It would be useful for readers interested in applying this methodology in other areas.

It would be useful to know how the PP from the model is computed: integrating down till the bottom, considering thee MLD?

**6. Discussion6.1 Trade-offs between different observations for parameter optimization**

In this section Authors use a number of times the following terms *poorly constrained*, *weakly constrained* and *unconstrained*, *un-optimized fields*, *not optimized but well defined*. Some definitions can be grasped from section 3.5. In my opinion it would make things more simple to have the formal definition of these terms and to know if, in same cases, they are equivalent/synonym.

Pg 21 lines 36-42 Although this cross-validation at different times and locations may give some indication of overfitting, it cannot determine whether the model reproduces observation through wrong mechanisms because a small misfit of cross-validation can be caused by missing validations of key variables or fluxes, e.g. ignorance of phytoplankton and PP in the experiment B, while a large misfit can be a result of the intrinsic heterogeneity of biological parameters in different times (Mattern et al., 2012) and locations (Kidston et al., 2011), e.g. underestimation of coastal surface chlorophyll in the experiment C.

In my opinion the sentence above is not very easy to follow, could Authors simplify?

Pg21 lines 54-55 On the other hand, counter examples exist where the 3D simulations outperform the 1D model (Hoshiba et al., 2018).

Could Authors explain better this sentence? Outperform with respect to what aspect?

---

## Referee Comment (RC2) · Peter Strutton (Referee) · 22 May 2020

This is a very useful contribution that explains the benefit that models can derive from the incorporation of satellite and BGC-Argo observations. The paper is timely and clearly written. I recommend publication after minor revisions.

Specific comments: The introduction is comprehensive. It could be shortened a bit (the 3rd and 5th paragraphs could mostly be removed) but this is not essential.

Methods switch between present and past tense. Also not a big deal, just disconcerting

for the reader.

P6 L48: Here and in subsequent equations/text I'm a bit confused. The float and satellite measure bbp700 and bbp670 respectively. So why are we now talking about bbp470? And what is meant by 'validated bbp470'?

P6 Eq 2 and 3: What are the units of the terms on the LHS? Please be more specific about what 'Phytoplankton' is. I think it's phytoplankton N.

P11 L69: Here and in section 3.1, the temporal resolution of the satellite data is not specified. I also think a bit more information here would be useful. How are monthly climatologies of the float profiles created? What distance from the 1D site is considered? Maybe this is described elsewhere and I missed it, but I see the other reviewer asked something similar.

P14-15: In the sub-section headings, it wouldn't hurt to remind us what experiments A, B and C are. That is 'satellite only' etc.

Figures 3 and 8: Why not just put the parameter labels on the y axes?

For the 3D case, I think it's correct to say that Figure 8 is an average of all model grid cells where the water depth is >1000m. That's a pretty big area. Yes, it's reasonably uniformly low chlorophyll, so one could make the case that this encompasses a contiguous bio-region. But why not choose a smaller box in the middle of the deep part of the basin, and perhaps one from the shelf, to illustrate the model performance? I suspect the latter will not perform as well, but that would still be interesting to know.

The results proceed in a logical fashion through the different experiments, 1D and 3D. The presentation of the results is clear and concise.

In a paper like this, readers will likely be looking for a clear recommendation: Which of the three options should they choose? I think the conclusions do a good job of summarising the recommendations and the figures represent what's lost if it's not possible to implement the best case scenario.

---

## Referee Comment (RC3) · Mara Freilich (Referee) · 26 May 2020

This manuscript presents a very well developed and well described parameter optimization process for biogeochemical models using available sea surface and profile observations. The authors show that a model parameterized with both surface chlorophyll from satellite and BGC-Argo profiles of chlorophyll and POC best represents the available observations of ecosystem state and fluxes. The authors demonstrate that the parameter choice has important implications for carbon cycling and export.

Comments:

2. Study Region

This section is very useful as an orientation to the region. Since one of the major objectives of the study is to analyze carbon export, it would be useful to include more in this section about what is known about carbon export in the Gulf of Mexico.

3. Methods

p. 6, Line 147: Earlier in this section, the satellite chlorophyll was adjusted to the float chlorophyll, but here the float backscattering is adjusted to the satellite backscattering. Why is this? These adjustments may have important implications for the POC profiles and partitioning of POC between phytoplankton, zooplankton, and POC. The reasoning for and implications of this choice should be explained.

p. 7, Line 167: What is the range of the vertical resolution of the ROMS model in the upper 200 meters? This information will be useful for comparison with the 1D model.

p. 8, Line 187: Since the biogeochemical model functional forms are essential to evaluating the model performance and the parameter optimization results, the biogeochemical model equations should be reproduced in this manuscript or in an appendix, rather than referring the reader to a different paper.

p. 8, Line 199: "Zooplankton and small detritus were assumed to amount to10% of phytoplankton biomass and the remaining fractions of POC attributed to large detritus." Why is this assumption made? Is this assumption only employed to define the initial condition?

p. 8, Line 210: The 1D model setup is sensible, but more information is needed to understand, evaluate, and reproduce the 1D model - Why is a 5 meter vertical resolution chosen in 1D and is the model sensitive to this choice? - What are the units for the diffusion coefficient and why are these values chosen? They are substantially higher than typical vertical diffusion coefficients in 3D models. - Are all of the parameters that

are taken from the 3D model seasonally varying (the mixed layer depth, temperature, solar radiation, and NO3 below 100 meters)? If so, what the temporal frequency at which they are updated? - Why is the temperature and mixed layer depth determined using values from the 3D model rather than the floats, which also have that data?

p. 11, Line 281: Why is the ratio of 0.1 between the sinking speed of the small and large detritus selected? How sensitive are the results to this choice?

4. Optimization of 1D model

This section is well written and clear. The discussion of the differences between the surface chlorophyll and vertical profiles is particularly clear and interesting.

p. 13, Line 329: "Unlike phytoplankton, the observations show that the POC concentrations are 19 mg C m-3 at about 200 m depth because of the existence of detritus (Figure 4c)." What is the evidence that the POC is in the detritus class rather than zooplankton? This point is not supported, but is used later to discriminate between models.

Section 4.2: In the section for each experiment, remind the reader what data is used the optimization and which parameters are included in the parameter optimization

Could you plot the mixed layer depth of the 1D model for comparison to the DCM depth? In the 1D model, the mixed layer depth is the main physical control. One option would be to plot the mixed layer depth in figure 3e.

The supplemental figure S1 shows that even the corrected satellite data does not capture the seasonality observed by the floats. One sentence here describing the differences that remain between the floats and corrected OC-CCI could help us to better assess why experiment A in particular does not capture the seasonality.

p. 15, Line 375: "Although a slight increase in the misfit occurs for the surface chlorophyll (∼5%)," Is the increase of 5% relative to experiment B?

Table 2: In the caption, explain what the dashes in the table mean. In the cases where the parameters are not included in the optimization, are the values that are presented in table 1 used? It would be helpful to the reader if the experiment that is discussed in the text is highlighted. Since different parameters are used in the 3D experiment, include additional lines in this table for the parameters that are used in the 3D experiment.

Figure 3: Include what data is used for each experiment in the figure caption. Could error bars be added to the observational points in this figure?

Figure 4: What do the error bars represent? Are they the interquartile range?

5. 3D biogeochemical model

p. 16, Line 421: How were the parameters that were modified manually chosen? It would be useful to provide more discussion of this choice.

p. 17, Line 442: Here the authors point to specific parameters that were inappropriate, but on p. 10, Line 246 the authors say that parameters are not allowed to exit a predefined range (which is shown in table 1). This seems inconsistent given that the method could have excluded inconsistent values. Could this difference be explained in more detail?

Section 5.1: The authors state that the 3D model does not perform well in coastal regions and therefore choose to exclude those regions from the model evaluation. This may be justified based on the statement in section 2 that there is little cross-shelf exchange in the Gulf of Mexico. However, this point should be discussed in more detail in order to justify ignoring the shelf. In particular, it is important to discuss the extent to which nutrients are supplied to the oligotrophic regions from either the shelf or the open ocean boundary. What is the importance of the boundaries relative to the biogeochemical cycling in the interior? In a 3D model, the boundaries could be very important for setting the primary production in the oligotrophic region.

p. 18, Line 451: What is meant by "different spatio-temporal scales between the two

model versions"? This point seems important and could be clarified. Is it referring to time stepping, resolution, retention in a 1D location, or the presence of a seasonal cycle?

p. 19, Line 479: "cannot" should be "can not"

---

## Author Comment (AC1) · 10 Jun 2020

**Responses to Reviewer 1**

We thank the reviewer for the constructive comments and suggestions which will be very helpful as we revise the manuscript. Below the complete reviewer comments are shown along with detailed responses to each comment (reviewer comments in black, responses in blue font)

**Review:**

Dear Editor, I read with great interest this manuscript related to the application of an optimization procedure combining satellite and BGC-Argo floats measurements with a biogeochemical model of the Gulf of Mexico. The manuscript is well organized and provides important insight to improve model parameterizations combining state-of the-art sensors data-streams. I found particularly important the results showing the importance of combining chlorophyll and bbp measured by BGCArgo to optimize the model in terms of POC and export. In my opinion it is also important the results that, to a large extent, the optimization procedure carried out on the 1D model can be translated to the 3D model implementation. Therefore, I suggest the publication of this manuscript after minor revisions reported below. My comments are indented in blue.

Minor revisions

**Response:** We would like to thank the reviewer for the positive assessment and constructive comments.

**1. Introduction**
Pg2 line 45. because the number of parameters increases exponentially with the number of state variables (Denman, 2003).

I would expect a polynomial [quadratic] not an exponential increase of the parameter number vs state variable number. In fact, Denman (2003) provides a polynomial formula for the increase of fluxes vs state variables and elaborates the estimate of increase of parameters basing on this formula.

**Response:** Thanks for noting this. We will revise as suggested.

Pg3 lines 68-70. This is especially problematic in oligotrophic regions where the maximum chlorophyll concentration (referred as the deep chlorophyll maximum, DCM) is pronounced near the base of the euphotic zone because of photo-acclimation (Cullen, 2015; Fennel and Boss, 2003).
Could Authors explain better what they mean? In general DCM can appear in mathematical terms even without photo-acclimation (e.g. Varela et al. 1994, Ryabov and Blasius 2008). I suggest Authors to be more specific on this point. Nonetheless, the effect of photo-acclimation mentioned here could produce an enhancement of the DCM feature [already explainable by mechanisms other than photo-acclimation] and could be particularly important for the specific area of the GoM.

References

Varela, R. A., A. Cruzado, and Joaquín Tintoré. "A simulation analysis of various biological and physical factors influencing the deep-chlorophyll maximum structure in oligotrophic areas." Journal of Marine Systems 5.2 (1994): 143-157.

Ryabov, Alexei B., and Bernd Blasius. "Population growth and persistence in a heterogeneous environment: the role of diffusion and advection." Mathematical Modelling of Natural Phenomena3.3 (2008): 42-86.

**Response:** Here we are referring to the fact that in oligotrophic regions where the deep chlorophyll maximum is pronounced and located relatively deep in the water column, satellite data of surface chlorophyll are insufficient for characterizing the vertically inhomogenous chlorophyll structure.

Furthermore, in general in the ocean, DCMs are maxima in chlorophyll but not biomass. Typically, there is no biomass maximum associated with the DCM, it is merely an artifact of photoacclimation acting on the exponentially declining light field. This is especially obvious in oligotrophic regions. While it might be true that a biomass maximum coincident with the DCM can be created in mathematical equations as the reviewer suggests (e.g. this is true in an NPDZ model that does not account for photoacclimation), this does not accurately reflect reality.

In our revised manuscript, we would like to make both of these points clearer.

**2. Study Region**

I would suggest Authors to add information about the recirculation times of the GoM waters with respect to the Atlantic Ocean boundary conditions (BC), this would give an idea on the relevance of the BC in the experiments.

**Response:** The recirculation times of the open Gulf is relatively long (certainly much longer than our simulation), but we are not aware of a study that specifically looks at this question. We feel that a discussion of recirculation times is well outside of the intended scope of our manuscript. Our treatment of the boundary conditions is consistent with previous modeling studies with the same or a slightly modified version of the same model (Xue et al. 2013, 2016, Yu et al. 2019), hence we are reasonably confident in our set-up.

Xue, Z., He, R., Fennel, K., Cai, W.-J., Lohrenz, S., Huang, W.-J., Tian, H., Ren, W., and Zang, Z.: Modeling $pCO_2$ variability in the Gulf of Mexico, Biogeosciences, 13, 4359-4377. 2016.

Xue, Z., He, R., Fennel, K., Cai, W.-J., Lohrenz, S., and Hopkinson, C., Modeling ocean circulation and biogeochemical variability in the Gulf of Mexico, Biogeosciences, 10, 7219-7234, doi:10.5194/bg-10-7219-2013. 2013.

Yu, L., Fennel, K., Wang, B., Laurent, A., Thompson, K. R., and Shay, L. K.: Evaluation of nonidentical versus identical twin approaches for observation impact assessments: an ensemble-Kalman-filter-based ocean assimilation application for the Gulf of Mexico, Ocean Science, 15, 1801–1814, https://doi.org/10.5194/os-15-1801-2019, 2019

Moreover it would be interesting to know what are the observed dominant plankton species, or plankton functional types, in the open-ocean part of GoM [pico-phytoplankton and flagellates?]. This could be useful to figure out the implications of the choice of considering one phytoplankton in the model when compared to data.

**Response:** We chose a one-phytoplankton model because it is simple and thus more likely that the model parameters can be constrained by the limited observations available. If we were using a more complex model, there would be more uncertain parameters making model optimization much more difficult. In addition, this relatively simple model has been successfully used to study the biogeochemical variabilities (Xue et al., 2013) and $CO_2$ air-sea fluxes (Xue et al., 2016) in the Gulf of Mexico. The same model has also been used for a number of studies of hypoxia generation in the northern Gulf of Mexico (e.g. Fennel et al. 2011, Yu et al. 2015, Laurent et al. 2017, 2018, Große et al. 2019). While we agree with the reviewer that analyses of model complexity are important and interesting, such a discussion is outside of the intended scope of this manuscript. There are dedicated publications on this (e.g. Kuhn et al. 2019). We believe we are making a novel contribution on a different important topic in this manuscript.

Fennel, K., Hetland, R., Feng, Y., and DiMarco, S.: A coupled physical-biological model of the Northern Gulf of Mexico shelf: model description, validation and analysis of phytoplankton variability, Biogeosciences, 8, 1881–1899, https://doi.org/10.5194/bg-8-1881-2011, 2011.

Große, F., Fennel, K., & Laurent, A.: Quantifying the relative importance of riverine and open-ocean nitrogen sources for hypoxia formation in the northern Gulf of Mexico. Journal of Geophysical Research: Oceans, 124. https://doi.org/10.1029/2019JC015230. 2019.

Kuhn, A. Fennel, K.: Evaluating ecosystem model complexity for the northwest North Atlantic through surrogate-based optimization, Ocean Modelling, https://doi.org/10.1016/j.ocemod.2019.101437, 2019.

Laurent, A., K. Fennel, W.-J. Cai,W.-J. Huang, L. Barbero, and R. Wanninkhof: Eutrophication- induced acidification of coastal waters in the northern Gulf of Mexico: Insights into origin and processes from a coupled physical-biogeochemical model, Geophys. Res. Lett., 44, 946–956, doi:10.1002/2016GL071881. 2017.

Laurent, A., Fennel, K., Ko, D. S., & Lehrter, J.: Climate change projected to exacerbate impacts of coastal eutrophication in the northern Gulf of Mexico. Journal of Geophysical Research: Oceans, 123, 3408–3426. https://doi.org/10.1002/2017JC013583. 2018.

Xue, Z., He, R., Fennel, K., Cai, W.-J., Lohrenz, S., and Hopkinson, C.: Modeling ocean circulation and biogeochemical variability in the Gulf of Mexico, Biogeosciences, 10, 7219–7234, https://doi.org/10.5194/bg-10-7219-2013, 2013.

Xue, Z., He, R., Fennel, K., Cai, W.-J., Lohrenz, S., Huang, W.-J., Tian, H., Ren, W., and Zang, Z.: Modeling pCO2 variability in the Gulf of Mexico, Biogeosciences, 13, 4359–4377, https://doi.org/10.5194/bg-13-4359-2016, 2016.

Yu, L., Fennel, K., Laurent, A., Murrell, M. C., and Lehrter, J. C.: Numerical analysis of the primary processes controlling oxygen dynamics on the Louisiana shelf, Biogeosciences, 12, 2063–2076, https://doi.org/10.5194/bg-12-2063-2015, 2015.

**3. Methods**
3.1. Biological observations
Pg 5 lines21-22 SeaWiFS (Sea-viewing Wide Field-of-view Sensor), MODIS (Moderate resolution Imaging Spectroradiometer), and MERIS (medium-spectral resolution imaging spectrometer) products. What about VIIRS? Is it included in the database?, please check.

**Response:** We checked and confirmed that indeed VIIRS is included in the database. We will correct it in our revised manuscript and thank the reviewer for pointing this out.

Pg 5 line 26 and particulate backscattering
Pg 5 lines 31-32 Satellite estimates were therefore corrected following the regression equation shown in Figure S2a (Figure S1c).
Pg 6 lines 45-48 The bbp700 from the floats is weakly correlated with the satellite estimates ($R^2=0.11$) and generally lower by a factor of ~0.45 than the satellite estimates (Figure S2b). The bbp700 profiles were therefore multiplied by 2.2 before being converted to bbp470 following the equ. 1.

In one case Authors consider the BGC-Argo the ground truth (Pg 5 lines 31-32) in the other Authors correct BGC-Argo with respect to satellite data (Pg 6 lines 45-48). The procedure seems a bit circular, could Authors explain better this part?
Pg 6 line 44 extent

**Response:** Chlorophyll and backscatter, regardless of whether they are obtained by satellite or in-situ sensors, are proxy measurements. That means they have to be converted and calibrated before they can be used. Here the BGC-Argo floats provided measurements of chlorophyll fluorescence and the volume scattering function at a centroid angle of $140^o$ and a wavelength of 700nm ($\beta(140^o, 700nm)$ m$^{-1}$ sr$^{-1}$), i.e. a measure of backscatter. The conversion from fluorescence into chlorophyll concentration was based on the sensor manufacturer's calibration. The conversion into bbp700 was performed by us following Green et al. (2014) and by cross-calibration against existing relationships with POC and phytoplankton biomass (Martinez-Vicente et al., 2013; Rasse et al., 2017). The resulting concentrations of phytoplankton biomass and POC as well as the ratio of chlorophyll to phytoplankton biomass are reasonable (please see figures 4 and 10). We will include a more in-depth explanation of this process in our revised manuscript.

Green, R. E., Bower, A. S. and Lugo-Fernandez, A.: First Autonomous Bio-Optical Profiling Float in the Gulf of Mexico Reveals Dynamic Biogeochemistry in Deep Waters, PLoS ONE, 9(7), 1–9, doi:10.1371/journal.pone.0101658, 2014.

Martinez-Vicente, V., Dall'Olmo, G., Tarran, G., Boss, E. and Sathyendranath, S.: Optical backscattering is correlated with phytoplankton carbon across the Atlantic Ocean, Geophysical Research Letters, 40, 1154–1158, doi:10.1002/grl.50252, 2013.

Rasse, R., Dall'Olmo, G., Graff, J., Westberry, T. K., van Dongen-Vogels, V. and Behrenfeld, M. J.: Evaluating Optical Proxies of Particulate Organic Carbon across the Surface Atlantic Ocean, Frontiers in Marine Science, 4(November), 1–18, doi:10.3389/fmars.2017.00367, 2017.

3.2. 3D Model description
Pg 7 lines 74-75 Medium-Range Weather Forecast ERA-Interim product with a horizontal resolution of 0.125o (ECMWF reanalysis, https://www.ecmwf.int/en/forecasts/datasets/reanalysisdatasets/era-interim).

The resolution reported in the link above for ERA-Interim is 80 km this does not match with the 0.125$^o$ (~12 km) reported in the manuscript, could Authors double check?

**Response:** The resolution of ~80 km is the model grid resolution used by the ERA-Interim's atmospheric model and reanalysis system. However, they also provide their dataset for download in higher (e.g. 0.125$^o$) or coarser resolutions (e.g. 3$^o$) which the original data are interpolated onto. We used the higher-resolution data set as stated in the manuscript.

3.3. 1D Model description
Pg8 line 10-12. The 1D model, which is similar to that used by Lagman et al. (2014) and Kuhn et al. (2015), covers the upper 200 m of the ocean with a vertical resolution of 5 m and is configured at one location in the central Gulf (see Figure 1).

Authors should add some comment about the choice of 5 m vertical resolution. What is the 3D model vertical resolution in the same region of the GoM? Why not taking 1m resolution that probably is the vertical resolution of data acquired by BGC-Argo floats?

**Response:** There are two main reasons why we chose 5 m. First, the vertical resolution in our BGC-Argo floats is 4-6 meters in the upper 200 m. Second, a 5-m vertical resolution in the 1D model keeps the computational cost reasonable. As stated in sections 3.4 and 3.5 of the manuscript, one parameter optimization experiment ran 36,000 model simulations (30 simulations/generation * 300 generations*4 replications). It took us about one day to run a single parameter optimization experiment and we performed 13 1D optimization experiments in total (A1-A4, B1-B4, C1-C5). This is in addition to all the 3D model runs we performed.

We are confident the vertical resolution is appropriate because we performed sensitivity tests using different vertical resolutions including 10 m, 5 m, 3 m, and 1 m in optimization C4. As shown in Figures 1 and 2 below, the choice of vertical resolution has little impact on model results except for minor jumps in the DCM depth (Figure 1e) and DCM magnitude (Figure 2) when a vertical resolution of 10 m is used.

The 3D model had 36 terrain-following sigma levels with refined resolution near the surface and bottom layers. The vertical resolution in the 3D model varied from a few meters near the surface to ~50 m near the depth of 200 m around the 1D site.

In our revised manuscript, we will add comments on the 1D model's vertical resolution as suggested.

[Figure]

Figure 1. The simulated annual cycle of surface chlorophyll (a), vertically integrated chlorophyll (b), vertically integrated phytoplankton (c), vertically integrated POC(d), and the depth (e) and magnitude (f) of the DCM by using different vertical resolutions.

[Figure]

Figure 2. The simulated vertical profiles of chlorophyll, phytoplankton, and POC by using different vertical resolutions.

Pg9 line 13-15. A higher diffusion coefficient ( $K_{ZI}$ =max($H^2$MLD /400,10))is applied in the turbulent surface layer and a lower diffusion coefficient ($K_{Z2}$=$K_{zI}$/2) is assigned to the quiescent bottom layer.

What is the unit of measure of K1,2, please add this information in the text. If it is m2/s it seems very high, because even with $H_{MLD}$ = 0 → $K_{ZI}$ =10 → $KZ2$ =5. Please, add also unit of measure of $H_{MLD}$, meters?

**Response:** The units of $K_{Z1}$ and $K_{Z2}$ are $m^2$/day. The unit of $H_{MLD}$ is in meters. We will add the units as suggested in our revised manuscript.

3.4. Parameter optimization method
Pg 11 line 63. and n is the number of base-test pairs including

I suggest to use consisting rather than including that in my opinion is confusing. In fact n accounts exactly to what specified (i.e. base –test pairs etc etc )_and not more.

**Response:** We will revise it as suggested.

The range of variability of Wphy spans many orders of magnitude and it is the most sensitive parameter, could Author comment on that?

**Response:** The range of sinking speeds for organic matter is large, because pico-plankton sink very slowly, if at all, while large detrital aggregates can sink very fast. We used the same range for all sinking speed in the optimization, as reported in Table 1 (for Wphy, WSdet, WLdet), although the speeds for Phy will be in the lower range of the bracket, speeds for SDet in the middle range, and speeds for Ldet in the upper range.

3.5. Parameter optimization experiments
Pg 11 line 69-71. For the parameter optimization of the 1D model, satellite chlorophyll within a 3´3 pixel (12 km´12 km area) around the 1D station and climatological monthly averages of the profiles from the bio-optical floats were used.

Did you considered all the BGC-Argo data available or only the ones near the virtual mooring of your 1D experiment? How you decided the BGC-Argo float to include in the optimization procedure?

**Response:** We used all BGC-Argo data by averaging all profiles into climatological monthly bins. We did this because 1) the BGC-Argo profiles were sparsely distributed in the Gulf of Mexico and there were few profiles around the 1D site, and 2) the deep Gulf of Mexico is quite homogenous horizontally.

In the revised manuscript, we will state this more clearly.

Pg 11 line 81-84. Prior tests have shown that the available observations cannot simultaneously constrain the sinking rates of small and large detritus (wSDet and wLDet)Therefore, a constant ratio of 0.1 between these two parameters (wSDet =0.1 wLDet) was imposed and only one of the two was optimized.

I suggest Authors to be clearer: I cannot get why the two parameters cannot be constrained. In general, after you completed your optimization procedure you could perturb the parameter 0.1 (ratio between WsDet and WlDet) to see if it corresponds to a minimum for your metrics F(p) or if there are better values other than 0.1. Or the system is unstable if you don't take 0.1? Please explain.

**Response:** These two parameters cannot be constrained simultaneously because they can compensate each other: an increase in one parameter can be counteracted by a decrease in the other. This is a well-known and much discussed issue in the parameter optimization literature. Basically, there is no information in the available observations that allows to distinguish between the two. Including both parameters in the optimization will degrade the model's predictive skill. To solve this problem, one can either introduce more independent observations or prior knowledge by, e.g. fixing these parameters to their prior values or defining a link between these parameters. Generally, and also in this case, observations are insufficient to

constrain all parameters and prior knowledge about the parameters has to be included. This is the common approach to address the underdetermination problem of parameter optimization in biological models.

In this study, the ratio of 0.1 was selected based on the prior values of the two parameters. Perturbing and estimating the ratio between two sinking velocities are equivalent to letting both parameters (wSDet and wLDet) into optimization and may degrade model's predictive skills. Without additional information about the depth distribution of Sdet and Ldet, which is not available, the ratio cannot be constrained.

The underdetermination problem and the practice of incorporating prior knowledges is also discussed on P20, L10-28 of our original manuscript.

Pg 12 line 92 please remove *and* from the equation.

**Response:** We will revise it as suggested.

**4. Optimization of 1D models**
General question: how you define the DCM depth in the presented analyses?

**Response:** In this study, the DCM depth was defined as the depth where the subsurface chlorophyll is at its maximum. We will include the definition of DCM depth in our revised manuscript.

4.1. Observations and base case
Pg 12. Lines 107-109. Unlike the surface chlorophyll, the vertically integrated chlorophyll as well as the phytoplankton and POC over the upper 200 m tend to be more constant with much less seasonality (Figure 3b-d).

This statement refers to observation? Please specify. Is it possible to add the error bar to the dots of Figure 3 as in the case of Figure 4?

**Response:** Yes, we are referring to observations which are represented by black dots. We will clarify this in our revised manuscript and add the error bar in Figure 3 as suggested.

Pg 12 line 14 in June and gradual shoaling after July (Figure 3e), reflecting the seasonality of the solar radiation.

With the term reflecting Authors mean that there is direct causality or correlation?

**Response:** We think that there is direct causality between the shoaling of DCM depth and solar radiation. In the oligotrophic regions, the DCM is strongly determined by photoacclimation (Cullen, 2015; Fennel and Boss, 2003). Previous studies based on BGC-Argo floats also suggest that the DCM is mainly light driven and located at the level of a particular isolume (e.g. Mignot et al., 2014; Xing et al., 2018). In our case, with the decrease of solar radiation after July, these isolumes and the DCM would therefore become shallower.

Cullen, J. J.: Subsurface Chlorophyll Maximum Layers : Enduring Enigma or Mystery Solved ?, Annual Review of Marine Science, 7, 207–239, doi:10.1146/annurev-marine-010213-135111, 2015.

Fennel, K. and Boss, E.: Subsurface maxima of phytoplankton and chlorophyll : Steady-state solutions from a simple model, Limnology and Oceanography, 48(4), 1521–1534, 2003.

Mignot, A., Claustre, H., Uitz, J., Poteau, A., D'Ortenzio, F. and Xing, X.: Understanding the seasonal dynamics of phytoplankton biomass and the deep chlorophyll maximum in oligotrophic environments: A Bio-Argo float investigation, Global Biogeochemical Cycles, 28(8), 1–21, doi:10.1002/2013GB004781, 2014.

Xing, X., Qiu, G., Boss, E., & Wang, H. Temporal and vertical variations of particulate and dissolved optical properties in the South China Sea. Journal of Geophysical Research: Oceans, 124. https://doi.org/10.1029/2018JC014880. 2019.

Pg 13 line 18 However, it fails to reproduce the deepening of the DCM in June

This deepening is related to a physical process (change in some environmental regulating factor) or to a biogeochemical process?

**Response:** This deepening is a result of both physical and biological processes. Firstly, as stated in the reply to the last comment, the increase of solar radiation would make the DCM deeper. Secondly, as the water column becomes more stratified in summer, chlorophyll concentrations in the upper layer will be lower as a result of nutrient limitation, which in turn increases light penetration through the water column and thus causing the DCM to deepen.

4.2. Results of the optimizations
4.2.1 Model-data misfits
4.2.2 Experiment A
Pg 13 line 41-42 The optimal parameter sets (A4, B2, and C4), which are selected based on casespecific misfit from these three groups, will be used in subsequent analyses …

Why Authors decide to use B2 rather than B4 that show a smaller total misfit? Is it better to take a realization with better case-specific misfit or better total misfit?

**Response:** Firstly, experiments in group B represent the scenario where we only have chlorophyll observations. In that case, we would have no idea about the model misfit for other variables except chlorophyll or the total misfit. Secondly, the case-specific misfit was the criterion that the parameter optimization procedure used to choose the best half population while "killing" the other half. Hence, using B2 rather than B4 as the optimal parameter sets makes sense.

We discussed the choice of B2 in our original manuscript. In realistic parameter optimization applications where only chl and no other observation are available the B2 optimization would be chosen. Revealing this weakness due to insufficient observations can help us to better understand the additional value of BGC-Argo floats and provide some scientific guidance for future parameter optimization studies. This is really the point we are trying to make here.

4.2.3 Experiment B
Pg 14 line 47-49 However, the vertical structure of chlorophyll deteriorates relative to the base case (Figure 4a) because of inappropriate estimates of the initial slope ($\alpha$=0.0101; see table 2) and the maximum ratio of chlorophyll to carbon ($\theta$max=0.0191; see table 2).

If Authors can judge a-priori that the values for $\alpha$ and $\theta$max resulting from the optimization are not appropriate, why they didn't consider different parameter ranges in the optimization procedure from the beginning, excluding bad values?

**Response:** Firstly, we have no prior knowledge of these parameters in the Gulf of Mexico and that is why we run parameter optimization to estimate their values. Our comments on these two parameters here were based on the results of our parameter optimization experiments.

Secondly, we feel these two parameters were inappropriate after we compared model results with vertical profiles of chlorophyll and phytoplankton. Since experiment A was designed to represent the scenario where only satellite observations are available, adding the information obtained from the BGC-Argo floats data into experiment A would not be a fair comparison to the other experiments and violate our assumptions of the experiment A.

Pg 14 lines 57-59. A straightforward interpretation is that the addition of subsurface observations reduces the model's degrees of freedom to fit one single observation type (surface chlorophyll).

Does this imply that a model with more parameters (e.g. more phytoplankton species) would fit better?

**Response:** Here we mean that the addition of subsurface observations yields more constraints on our biological model because the model should fit both surface and subsurface observations simultaneously.

With respect to reviewer's comment, it tends to be true but not always. However, the better fit is often the result of overfitting rather than a true improvement in predictive skill. It is of utmost importance to avoid overfitting. The increased number of parameters in more complex models may increase the risk of degrading model's predictive skills.

Pg15 line 70 In contrast to the observations where detritus dominates POC …

What observation? Please add reference.

**Response:** Here we referred to our BGC-Argo floats shown in Figure 4. We will include reference in our revised manuscript as suggested.

4.2.4 Experiment C
Pg 15 lines 76-78 As shown in Figure 4a, the annually averaged depth of DCM of 80 m coincides with the observed DCM, primarily because experiment reproduces the deepening of the DCM in summer.

Interesting. Can Authors explain if there is a specific parameter/mechanism that controls this dynamical deepening of the DCM? Or it is a complex combination of parameters values generating this emergent property?

**Response:** As we mentioned above, the deepening of the DCM in the summer resulted from a combination of increased solar radiation and decreased light attenuation in the upper layer. Since the different experiments here used the same inputs of solar radiation, the deepening of the DCM in experiment C must be a result of changes in parameter values which can influence chlorophyll concentrations and hence light attenuation in the upper layer. To illustrate this, we performed sensitivity simulations where we used the optimal parameters from experiment C but set each optimized parameter, one at a time, to its prior value. As shown in Figure 3, the optimization induced changes in large detritus remineralization rate ($r_{LD}$), half-saturation for NH4 uptake ($K_{NH4}$), and large detritus sinking velocity ($w_{LDet}$) can have large impacts on chlorophyll concentrations and the depth of DCM.

[Figure]

Figure 3: Vertical distributions of simulated chlorophyll from different sensitivity tests.

5. 3D biogeochemical model
Could Author explain better the manual correction described at pg 17 lines 24-25? If they consider the corrected values more realistic why they didn't narrowed the parameter variability range in the optimization experiment? Or the 1D vs 3D implementations do not allow this?

For example, is there a simple explanation for the need to set KNH4 to 0.01 in the 3D experiments? It would be useful for readers interested in applying this methodology in other areas.

**Response:** We did manual modifications because when the resulting parameters were directly applied, the model-data agreement in 3D models was not as well as in 1D models in some respects, but the most important features were well preserved. In experiment C, chlorophyll concentrations in the upper layer of 3D model were lower than the 1D model and farther away from observations. This might be a result of differences between 1D and 3D models and has been also reported in other studies (e.g. Kane et al., 2011; Hoshiba et al., 2018). However, the depth of the DCM and the non-zero POC concentrations at 200 m with appropriate contributions from each component were well preserved. We therefore did some tests manually by tuning one or two parameters and final set the KNH4 to 0.01 in order to have a better agreement with respect to observations. All of these modifications were based on our tests.

In our case, although the 1D parameter sets could not be used in 3D models directly, they were sufficient to reproduce the main features as in 1D models and largely simplified the following subjective tuning of 3D models by limiting the number of parameters to be adjusted. We will include some discussion in our revised manuscript.

Kane, A., Moulin, C., Thiria, S., Bopp, L., Berrada, M., Tagliabue, A., Crépon, M., Aumont, O. and Badran, F.: Improving the parameters of a global ocean biogeochemical model via variational assimilation of in situ data at five time series stations, Journal of Geophysical Research: Oceans, 116, 1–14, doi:10.1029/2009JC006005, 2011.

Hoshiba, Y., Hirata, T., Shigemitsu, M., Nakano, H., Hashioka, T., Masuda, Y. and Yamanaka, Y.: Biological data assimilation for parameter estimation of a phytoplankton functional type model for the western North Pacific, Ocean Science, 14, 371–386, doi:10.5194/os-14-371-2018, 2018.

It would be useful to know how the PP from the model is computed: integrating down till the bottom, considering thee MLD?

**Response:** The primary production in the 3D models was integrated down to 200 m, consistent with the primary production obtained from the 1D models. There is not enough available light below 200 m for photosynthesis to occur, therefore cutting off at 200 m yields the same values of integrated primary production as integration down to the seafloor would. As suggested, we will state this in our revised manuscript.

6. Discussion
6.1 Trade-offs between different observations for parameter optimization
In this section Authors use a number of times the following terms poorly constrained, weakly constrained and unconstrained, un-optimized fields, not optimized but well defined. Some definitions can be grasped from section 3.5. In my opinion it would make things more simple to have the formal definition of these terms and to know if, in same cases, they are equivalent/synonym.

**Response:** The poorly constrained and weakly constrained parameters are synonymous, meaning that they cannot be constrained in confidence. In our study, it can be simply understood that their estimates were inappropriate. The unconstrained and un-optimized fields are equivalent meaning that they were not included in the parameter optimization procedure. The not-optimized but well-defined parameters are those parameters which were not included in the parameter optimization but their prior values were coincidently reasonable. As suggested, we will try to simplify the language and include these definitions in our revised manuscript.

Pg 21 lines 36-42 Although this cross-validation at different times and locations may give some indication of overfitting, it cannot determine whether the model reproduces observation through wrong mechanisms because a small misfit of cross-validation can be caused by missing validations of key variables or fluxes, e.g. ignorance of phytoplankton and PP in the experiment B, while a large misfit can be a result of the intrinsic heterogeneity of biological parameters in different times (Mattern et al., 2012) and locations (Kidston et al., 2011), e.g. underestimation of coastal surface chlorophyll in the experiment C.

In my opinion the sentence above is not very easy to follow, could Authors simplify?

**Response:** As suggested, we will revise these sentences into:

*However, even when cross-validation at different times and locations produces large misfits, we cannot conclude that the models reproduce observations through wrong mechanisms. This is because the large misfit can be a result of intrinsic heterogeneity of biological parameters at different times (Mattern et al., 2012) and locations (Kidston et al., 2011).*

Pg21 lines 54-55 On the other hand, counter examples exist where the 3D simulations outperform the 1D model (Hoshiba et al., 2018).

Could Authors explain better this sentence? Outperform with respect to what aspect?

**Response:** Here we mean that applying parameters obtained from 1D models could result in worse or better fitness in 3D models. For instance, Hoshiba et al. 2018 performed parameter optimization by 1D models and applied the resulting parameters in 3D models. As a result, the 3D model had better fitness than the 1D model with respect to vertical profiles of phytoplankton and nitrate (please see their section 3.4 and Figure 10a,b). As suggested, we will clarify this in our revised manuscript.

Hoshiba, Y., Hirata, T., Shigemitsu, M., Nakano, H., Hashioka, T., Masuda, Y. and Yamanaka, Y.: Biological data assimilation for parameter estimation of a phytoplankton functional type model for the western North Pacific, Ocean Science, 14, 371–386, doi:10.5194/os-14-371-2018, 2018.

---

## Author Comment (AC2) · 10 Jun 2020

**Response to Peter Strutton (Referee #2)**

We thank the reviewer for the constructive comments and suggestions which will be very helpful as we revise the manuscript. Below the complete reviewer comments are shown along with detailed responses to each comment (reviewer comments in black, responses in blue font)

**Review:**

This is a very useful contribution that explains the benefit that models can derive from the incorporation of satellite and BGC-Argo observations. The paper is timely and clearly written. I recommend publication after minor revisions.

**Response:** We would like to thank the reviewer for the positive assessment and constructive comments.

**Specific comments:**

The introduction is comprehensive. It could be shortened a bit (the 3rd and 5th paragraphs could mostly be removed) but this is not essential.

**Response**: Thanks for these suggestions. The 5th paragraph is to show weakness of satellite observations which served as the motivation of this study and is highly related to some main conclusions. We will consider shortening this paragraph, but we would like to keep it in our revised manuscript. As suggested, the 3rd paragraph will be removed.

Methods switch between present and past tense. Also not a big deal, just disconcerting for the reader.

**Response:** We will look at this carefully to make sure the tense is used appropriately.

P6 L48: Here and in subsequent equations/text I'm a bit confused. The float and satellite measure bbp700 and bbp670 respectively. So why are we now talking about bbp470? And what is meant by 'validated bbp470'?

**Response:** We talked about bbp470 because the empirical relationship that we used to estimate phytoplankton and POC was based on bbp470 (please see equ. 2-3). We had to convert measurements of bbp700 and bbp670 to bbp470 based on the equ. 1 before we estimated phytoplankton and POC.

We will clarify this and make it clearer in our revised manuscript.

P6 Eq 2 and 3: What are the units of the terms on the LHS? Please be more specific about what 'Phytoplankton' is. I think it's phytoplankton N.

**Response:** Yes, the phytoplankton and POC were in unit of mmol N $m^{-3}$. We actually have mentioned it in P6-7 L58-60 in our original manuscript. We will also revise and make it clearer here as suggested.

P11 L69: Here and in section 3.1, the temporal resolution of the satellite data is not specified. I also think a bit more information here would be useful. How are monthly climatologies of the float profiles created? What distance from the 1D site is considered? Maybe this is described elsewhere and I missed it, but I see the other reviewer asked something similar.

**Response:** In this study, we used monthly satellite estimates of surface chlorophyll in the parameter optimization. The monthly climatology of float profiles was created by averaging all profiles collected in the Gulf of Mexico into monthly bins. We did this because 1) the BGC-Argo float profiles were sparsely distributed in the Gulf of Mexico and there were insufficient profiles around the 1D site (please see figure 1 in our original manuscript), and 2) the deep ocean part of the Gulf of Mexico is quite homogenous horizontally. As suggested, we will include some description and explanation in our revised manuscript.

P14-15: In the sub-section headings, it wouldn't hurt to remind us what experiments A, B and C are. That is 'satellite only' etc.

**Response:** Agree. We will revise it as suggested

Figures 3 and 8: Why not just put the parameter labels on the y axes?

**Response:** Yes, will do as suggested.

For the 3D case, I think it's correct to say that Figure 8 is an average of all model grid cells where the water depth is >1000m. That's a pretty big area. Yes, it's reasonably uniformly low chlorophyll, so one could make the case that this encompasses a contiguous bio-region. But why not choose a smaller box in the middle of the deep part of the basin, and perhaps one from the shelf, to illustrate the model performance? I suspect the latter will not perform as well, but that would still be interesting to know.

**Response:** To the first point, since the BGC-Argo float profiles are sparsely distributed in the deep part of the Gulf of Mexico (please see Figure 1 in our original manuscript), choosing a smaller box would mean that much fewer float profiles are available for the model-data comparison. Also, since the region is quite homogenous horizontally, we feel it is appropriate to average over all. In Figure 4, we show the interquartile range of the profiles in space and time (black bars). These are very similar if calculated only for July. Thus, the error bars in Figure 4 give a good indication of the spatial variability. We will add this in the revised manuscript.

To the second point, the BGC-Argo floats were deployed only in the deep part of the Gulf, hence no float profiles are available on the shelf. However, we agree that it would be interesting to show the validation in smaller boxes from the deep ocean and shelf, and will include this comparison with satellite surface chlorophyll in the supplementary of our revised manuscript.

The results proceed in a logical fashion through the different experiments, 1D and 3D. The presentation of the results is clear and concise.

**Response:** Thank you, we really appreciate this comment.

In a paper like this, readers will likely be looking for a clear recommendation: Which of the three options should they choose? I think the conclusions do a good job of summarizing the recommendations and the figures represent what's lost if it's not possible to implement the best case scenario.

**Response:** Thank you, we are happy to hear this.

---

## Author Comment (AC3) · 10 Jun 2020

**Responses to Mara Freilich (Reviewer 3)**

We thank the reviewer for the constructive comments and suggestions which will be very helpful as we revise the manuscript. Below the complete reviewer comments are shown along with detailed responses to each comment (reviewer comments in black, responses in blue font)

**Review:**

This manuscript presents a very well developed and well described parameter optimization process for biogeochemical models using available sea surface and profile observations. The authors show that a model parameterized with both surface chlorophyll from satellite and BGC-Argo profiles of chlorophyll and POC best represents the available observations of ecosystem state and fluxes. The authors demonstrate that the parameter choice has important implications for carbon cycling and export.

Comments:
2. Study Region
This section is very useful as an orientation to the region. Since one of the major objectives of the study is to analyze carbon export, it would be useful to include more in this section about what is known about carbon export in the Gulf of Mexico.

**Response:** We may include some key references in our revision, but would like to note that the major objective of this study is to evaluate the trade-offs between different observation types for biological model validation and parameter optimization. Our model-data comparison of carbon export is to illustrate the model performance and additional value brought by BGC-Argo profiles to parameter optimization. Studying the carbon export in the Gulf of Mexico is not within the intended scope of this study.

3. Methods
p. 6, Line 147: Earlier in this section, the satellite chlorophyll was adjusted to the float chlorophyll, but here the float backscattering is adjusted to the satellite backscattering. Why is this? These adjustments may have important implications for the POC profiles and partitioning of POC between phytoplankton, zooplankton, and POC. The reasoning for and implications of this choice should be explained.

**Response:** Chlorophyll and backscatter, regardless of whether they are obtained by satellite or in-situ sensors, are proxy measurements. That means they have to be converted and calibrated before they can be used. Here the BGC-Argo floats provided measurements of chlorophyll fluorescence and the volume scattering function at a centroid angle of $140^o$ and a wavelength of $700nm$ ($\beta(140^o, 700nm)$ m-1 sr-1), i.e. a measure of backscatter. The conversion from fluorescence into chlorophyll concentration was based on the sensor manufacturer's calibration. The conversion into bbp700 was performed by us following Green et al. (2014) and by cross-calibration against existing relationships with POC and phytoplankton biomass (Martinez-Vicente et al., 2013; Rasse et al., 2017). The resulting concentrations of phytoplankton biomass and POC as well as the ratio of chlorophyll to phytoplankton biomass are reasonable (please see figures 4 and 10). We will include a more in-depth explanation of this process in our revised manuscript.

Since this comment is similar to comments by Reviewer #1 under 3.1, we point also to those responses.

Green, R. E., Bower, A. S. and Lugo-Fernandez, A.: First Autonomous Bio-Optical Profiling Float in the Gulf of Mexico Reveals Dynamic Biogeochemistry in Deep Waters, PLoS ONE, 9(7), 1–9, doi:10.1371/journal.pone.0101658, 2014.

Martinez-Vicente, V., Dall'Olmo, G., Tarran, G., Boss, E. and Sathyendranath, S.: Optical backscattering is correlated with phytoplankton carbon across the Atlantic Ocean, Geophysical Research Letters, 40,

1154–1158, doi:10.1002/grl.50252, 2013.

Rasse, R., Dall'Olmo, G., Graff, J., Westberry, T. K., van Dongen-Vogels, V. and Behrenfeld, M. J.: Evaluating Optical Proxies of Particulate Organic Carbon across the Surface Atlantic Ocean, Frontiers in Marine Science, 4(November), 1–18, doi:10.3389/fmars.2017.00367, 2017.

p. 7, Line 167: What is the range of the vertical resolution of the ROMS model in the upper 200 meters? This information will be useful for comparison with the 1D model.

**Response**: The 3D model had 36 terrain-following sigma levels which means that the vertical resolution varies with the bathymetry. The vertical resolution in the 3D model ranges from a few meters near the surface to ~50 m near the depth of 200 m around the 1D site. We will add this during the revision.

We also point to our response to comments by Reviewer #1 under 3.2.

p. 8, Line 187: Since the biogeochemical model functional forms are essential to evaluating the model performance and the parameter optimization results, the biogeochemical model equations should be reproduced in this manuscript or in an appendix, rather than referring the reader to a different paper.

**Response:** The model code is published and freely available and the model equations, originally published in 2006, have recently been republished in the Appendix of Laurent et al. (2017). The model has been used in well over a dozen publications by our group and by hundreds of other researchers (the model is part of the widely used ROMS distribution). We do not agree that it is necessary or frankly appropriate to republish the same equations which each manuscript. This is also not common for other widely used models. We will make a specific reference to the equation in the Appendix of Laurent et al. (2017) when we revise.

Laurent, A., K. Fennel, W.-J. Cai, W.-J. Huang, L. Barbero, and R. Wanninkhof (2017), Eutrophication-induced acidification of coastal waters in the northern Gulf of Mexico: Insights into origin and processes from a coupled physical-biogeochemical model, Geophys. Res. Lett., 44, 946 – 956, doi:10.1002/2016GL071881.

p. 8, Line 199: "Zooplankton and small detritus were assumed to amount to 10% of phytoplankton biomass and the remaining fractions of POC attributed to large detritus." Why is this assumption made? Is this assumption only employed to define the initial condition?

**Response:** This assumption is also used for the open boundary conditions. For the biological model, the open boundary conditions are usually unavailable. Here we assumed the zooplankton and small detritus accounted for 10% of phytoplankton following Gomez et al., 2018 in which 20% was assumed for a different biological model in the Gulf of Mexico. We also do some sensitivity tests in the first year (2010) by perturbing this fraction to be 5%, 20%, and 40%. As shown in Figure 1, the choice of this fraction has little impact on the model results, e.g. phytoplankton, zooplankton, and POC. In the revised manuscript, we will add a short explanation of this assumption.

Gomez, F. A., Lee, S.-K., Liu, Y., Hernandez Jr., F. J., Muller-Karger, F. E., and Lamkin, J. T.: Seasonal patterns in phytoplankton biomass across the northern and deep Gulf of Mexico: a numerical model study, Biogeosciences, 15, 3561–3576, https://doi.org/10.5194/bg-15-3561-2018, 2018

[Figure]

Figure 1. The simulated vertical profiles of phytoplankton, zooplankton, and POC from different sensitivity tests. The solid lines represent median profiles while the dash lines represent interquartile range.

p. 8, Line 210: The 1D model setup is sensible, but more information is needed to understand, evaluate, and reproduce the 1D model - Why is a 5 meter vertical resolution chosen in 1D and is the model sensitive to this choice? - What are the units for the diffusion coefficient and why are these values chosen? They are substantially higher than typical vertical diffusion coefficients in 3D models. - Are all of the parameters that are taken from the 3D model seasonally varying (the mixed layer depth, temperature, solar radiation, and NO3 below 100 meters)? If so, what the temporal frequency at which they are updated? - Why is the temperature and mixed layer depth determined using values from the 3D model rather than the floats, which also have that data?

**Response:**
There are two main reasons why we chose 5 m. First, the vertical resolution in our BGC-Argo floats is 4-6 meters in the upper 200 m. Second, a 5-m vertical resolution in the 1D model keeps the computational cost reasonable. As stated in sections 3.4 and 3.5 of the manuscript, one parameter optimization experiment ran 36,000 model simulations (30 simulations/generation * 300 generations*4 replications). It took us about one day to run a single parameter optimization experiment and we performed 13 1D optimization experiments in total (A1-A4, B1-B4, C1-C5). This is in addition to all the 3D model runs we performed.

We are confident the vertical resolution is appropriate because we performed tests of the effect by using different vertical resolutions including 10 m, 5 m, 3 m, and 1 m for optimization C4. As shown in the Figure 2 and 3 below, the choice of vertical resolution has little impact on model results except for some small jumps in the DCM depth (Figure 2e) and DCM magnitude (Figure 2) when the vertical resolution of 10 m is used.

The unit of the diffusion coefficient is $m^2\ day^{-1}$ and its value was tuned such that our 1D and 3D models behave similarly. We will add the unit of diffusion coefficient in our revised manuscript.

Finally, the physical input was obtained from daily output of the 3D model. Float data were not used as physical inputs because of their low temporal resolution (~2 weeks).

We also point to our responses to comments by Reviewer #1 under 3.2.

[Figure]

Figure 2. The simulated annual cycle of surface chlorophyll (a), vertically integrated chlorophyll (b), vertically integrated phytoplankton (c), vertically integrated POC(d), and the depth (e) and magnitude (f) of the DCM by using different vertical resolutions.

[Figure]

Figure 3. The simulated (colored lines) vertical profiles of chlorophyll, phytoplankton, and POC by using different vertical resolutions.

p. 11, Line 281: Why is the ratio of 0.1 between the sinking speed of the small and large detritus selected? How sensitive are the results to this choice?

**Response:** We refer to our response to Reviewer #1, comment on Pg 11 line 81-84, which we paste again here for sake of convenience:

These two parameters cannot be constrained simultaneously because they can compensate each other: an increase in the one parameter can be counteracted by a decrease in the other. This is a well-known and much discussed issue in the parameter optimization literature. Basically, there is no information in the

available observations that allows to distinguish between the two. Including both parameters in the optimization will degrade the model's predictive skill. To solve this problem, one can either introduce more independent observations or prior knowledge by, e.g. fixing these parameters to their prior values or defining a link between these parameters. Generally, and also in this case, observations are insufficient to constrain all parameters and prior knowledge about the parameters has to be included. This is the common approach to addressing the underdetermination problem of parameter optimization in biological models.

In this study, the ratio of 0.1 was selected based on the prior values of the two parameters. Perturbing and estimating the ratio between two sinking velocities are equivalent to letting both parameters (wSDet and wLDet) into optimization and may degrade model's predictive skills. Without additional information about the depth distribution of Sdet and Ldet, which is not available, the ratio cannot be constrained.

The underdetermination problem and the practice of incorporating prior knowledges is also discussed on P20, L10-28 of our original manuscript.

4. Optimization of 1D model
This section is well written and clear. The discussion of the differences between the surface chlorophyll and vertical profiles is particularly clear and interesting.

**Response:** Thank you for this comments.

p. 13, Line 329: "Unlike phytoplankton, the observations show that the POC concentrations are 19 mg C m-3 at about 200 m depth because of the existence of detritus (Figure 4c)." What is the evidence that the POC is in the detritus class rather than zooplankton? This point is not supported, but is used later to discriminate between models.

**Response:** We would like to revise this as follows: *"… the POC concentrations are 19 mg C m-3 at about 200 m depth because of the existence of detritus, or zooplankton, or both"*. We would also like to revise on P15 L70-71 of original manuscript to *"In contrast to the observations where the phytoplankton's contribution is neglectable, the simulated POC at 200 m is dominated by phytoplankton (49%)"*.

Section 4.2: In the section for each experiment, remind the reader what data is used the optimization and which parameters are included in the parameter optimization

**Response:** As stated already in response to Reviewer 2, we will revise as suggested.

Could you plot the mixed layer depth of the 1D model for comparison to the DCM depth? In the 1D model, the mixed layer depth is the main physical control. One option would be to plot the mixed layer depth in figure 3e.

**Response:** Since the Figure 3e will be busy with a plot of the mixed layer, we will plot the seasonal mean levels of the mixed layer in Figure 4 and hope that will convey the information this reviewer is looking for.

The supplemental figure S1 shows that even the corrected satellite data does not capture the seasonality observed by the floats. One sentence here describing the differences that remain between the floats and corrected OC-CCI could help us to better assess why experiment A in particular does not capture the seasonality.

**Response:** In the experiment A, the satellite estimates of surface chlorophyll was used in parameter optimization and the model results were compared with the satellite data in the Figure 3a. The differences between the satellite and float data would not be a reason for experiment A not capturing the peak of satellite surface chlorophyll. It could be a result of many factors and might be improved by assigning a higher weight on the peaks of surface chlorophyll. However, studying the peaks of surface chlorophyll was not the purpose

in this study. Nonetheless, the parameter optimization resulted in a large improvement in surface chlorophyll in the experiment A.

p. 15, Line 375: "Although a slight increase in the misfit occurs for the surface chlorophyll (_5%)," Is the increase of 5% relative to experiment B?

**Response:** The slight increase here was relative to the base case. The full sentence in the original manuscript is *"Although a slight increase in the misfit occurs for the surface chlorophyll (~5%), the total misfit is reduced by 75% compared to the base case."* We hope this is clear.

Table 2: In the caption, explain what the dashes in the table mean. In the cases where the parameters are not included in the optimization, are the values that are presented in table 1 used? It would be helpful to the reader if the experiment that is discussed in the text is highlighted. Since different parameters are used in the 3D experiment, include additional lines in this table for the parameters that are used in the 3D experiment.

**Response**: Thanks for this comment. We will revise as suggested.

Figure 3: Include what data is used for each experiment in the figure caption. Could error bars be added to the observational points in this figure?

**Response:** Yes, we will revise as suggested.

Figure 4: What do the error bars represent? Are they the interquartile range?
**Response:** Yes, they are interquartile range. We will state that explicitly.

5. 3D biogeochemical model
p. 16, Line 421: How were the parameters that were modified manually chosen? It would be useful to provide more discussion of this choice.

**Response:** Same as comment "5. 3D biogeochemilca model" by Reviewer 1 and same response. We paste it again here for sake of convenience:

We did manual modifications because when the resulting parameters were directly applied, the model-data agreement in 3D models was not as well as in 1D models in some aspects, but the most important features were well preserved. In experiment C, chlorophyll concentrations in the upper layer of 3D model were lower than the 1D model and farther away from observations. This might be a result of differences between 1D and 3D models and has been also reported in other studies (e.g. Kane et al., 2011; Hoshiba et al., 2018). However, the depth of the DCM and the non-zero POC concentrations at 200 m with appropriate contributions from each component were well preserved. We therefore did some tests manually by tuning one or two parameters and final set the KNH4 to 0.01 in order to have a better agreement with respect to observations. All of these modifications were based on our tests.

In our case, although the 1D parameter sets could not be used in 3D models directly, they were sufficient to reproduce the main features as in 1D models and largely simplified the following subjective tuning of 3D models by limiting the number of parameters to be adjusted. We will include some discussion in our revised manuscript.

p. 17, Line 442: Here the authors point to specific parameters that were inappropriate, but on p. 10, Line 246 the authors say that parameters are not allowed to exit a predefined range (which is shown in table 1). This seems inconsistent given that the method could have excluded inconsistent values. Could this difference be explained in more detail?

**Response:** We think there might be a misunderstanding here. The predefined range of parameters was a collection of their values that had been used in other studies, models, and ecosystem regimes. That means not all parameter values within this range are appropriate for our model. The parameter optimization is therefore to search for an optimal parameter set within this predefined range by minimizing the misfit between model and observations. However, the parameter optimization cannot constrain all parameters (e.g. producing the inappropriate estimates) because of insufficient observations, which is referred to as the underdetermination problem (see earlier responses here and for Reviewer 1). For instance, in our experiment B, the parameter optimization fitted our model to chlorophyll observations but sacrificed other aspects of the model's performance which were degraded, e.g. the vertical profiles of phytoplankton. That is why we said the estimates of the maximum ratio between chlorophyll and phytoplankton ($\theta max$= 0.0158) in experiment B was inappropriate. An appropriate estimate of this parameter requires more independent observations, e.g. of phytoplankton that we used in the experiment C.

Section 5.1: The authors state that the 3D model does not perform well in coastal regions and therefore choose to exclude those regions from the model evaluation. This may be justified based on the statement in section 2 that there is little cross-shelf exchange in the Gulf of Mexico. However, this point should be discussed in more detail in order to justify ignoring the shelf. In particular, it is important to discuss the extent to which nutrients are supplied to the oligotrophic regions from either the shelf or the open ocean boundary. What is the importance of the boundaries relative to the biogeochemical cycling in the interior? In a 3D model, the boundaries could be very important for setting the primary production in the oligotrophic region.

**Response:** Firstly, we excluded coastal regions because we have no float data in coastal regions used for parameter optimization and model validation. All parameter optimization experiments in this study were designed for modelling the deep ocean part of the Gulf of Mexico. We will clarify this in our revised manuscript.

Secondly, if we define the boundary between coastal regions and open ocean as 1,000 m, the DIN fluxes (NO3+NH4) across the 1,000 m isobath is a sink for the open ocean and nearly balanced by the inputs from open boundaries. Specifically, for the upper 200 m where the primary production occurs, the amount of DIN transported is $2.7 \times 10^{11}$ mol N yr$^{-1}$ (sink) from the open ocean into coastal region and $2.1 \times 10^{11}$ mol N yr$^{-1}$ (source) from open boundaries. The primary production in the deep part of the Gulf of Mexico is $10.6 \times 10^{11}$ mol N yr$^{-1}$ (sink for DIN) which is mainly supported by the vertical transport and interior nitrogen recycle in the upper 200 m.

p. 18, Line 451: What is meant by "different spatio-temporal scales between the two model versions"? This point seems important and could be clarified. Is it referring to time stepping, resolution, retention in a 1D location, or the presence of a seasonal cycle?

**Response:** The different spatial-temporal scales refer to a lot of factors here. The 1D model is configured in a single station while the 3D model covers the whole Gulf of Mexico. In the vertical direction, the 1D model only simulates the upper 200 m but the 3D model has the whole water column. For temporal scales, the 1D model is simulated for one year but the 3D model runs for 6 years from 2010 to 2016. The model steps are also different between the 1D and 3D models. As suggested, we will clarify this in our revised manuscript.

p. 19, Line 479: "cannot" should be "can not"

**Response:** We will revise as suggested.